# Perinatal Tissue-Derived Stem Cells: An Emerging Therapeutic Strategy for Challenging Neurodegenerative Diseases

**DOI:** 10.3390/ijms25020976

**Published:** 2024-01-12

**Authors:** Annalisa Bruno, Cristina Milillo, Federico Anaclerio, Carlotta Buccolini, Anastasia Dell’Elice, Ilaria Angilletta, Marco Gatta, Patrizia Ballerini, Ivana Antonucci

**Affiliations:** 1Center for Advanced Studies and Technology (CAST), “G. d’Annunzio” University of Chieti-Pescara, 66100 Chieti, Italy; a.bruno@unich.it (A.B.); cristina.milillo@unich.it (C.M.); carlottabuccolini@alice.it (C.B.); anastasia.dellelice@gmail.com (A.D.); i.antonucci@unich.it (I.A.); 2Department of Innovative Technologies in Medicine & Dentistry, “G. d’Annunzio” University of Chieti-Pescara, 66100 Chieti, Italy; 3Department of Psychological, Health and Territorial Sciences, “G. d’Annunzio” University of Chieti-Pescara, 66100 Chieti, Italy

**Keywords:** gestational stem cells, neurodegenerative diseases, Parkinson’s disease, Huntington’s disease, Alzheimer’s disease, amyotrophic lateral sclerosis, cell transplantation therapy

## Abstract

Over the past 20 years, stem cell therapy has been considered a promising option for treating numerous disorders, in particular, neurodegenerative disorders. Stem cells exert neuroprotective and neurodegenerative benefits through different mechanisms, such as the secretion of neurotrophic factors, cell replacement, the activation of endogenous stem cells, and decreased neuroinflammation. Several sources of stem cells have been proposed for transplantation and the restoration of damaged tissue. Over recent decades, intensive research has focused on gestational stem cells considered a novel resource for cell transplantation therapy. The present review provides an update on the recent preclinical/clinical applications of gestational stem cells for the treatment of protein-misfolding diseases including Alzheimer’s disease (AD), Parkinson’s disease (PD), Huntington’s disease (HD) and amyotrophic lateral sclerosis (ALS). However, further studies should be encouraged to translate this promising therapeutic approach into the clinical setting.

## 1. Introduction

Neurodegenerative diseases (NDDs) refer to a heterogeneous group of pathological conditions characterized by the loss of neurons in the brain or spinal cord usually associated with a wide spectrum of clinical presentation including cognitive, psychiatric, and motor deficits [1]. NDDs can be triggered by either acute or chronic events. Acute cases, such as ischemic stroke and traumatic brain or spinal cord injuries are caused by a traumatic event leading to the loss of neurons at the site of damage. On the other hand, chronic cases are linked to the more selective loss of cell populations like dopaminergic neurons in Parkinson’s disease (PD) and motor neurons in amyotrophic lateral sclerosis (ALS) [2,3,4,5]. Most NDDs are characterized by an accumulation of misfolded proteins (proteins with altered physicochemical properties), which represent a hallmark of the disease. Examples of protein accumulations are (i) a-Synuclein (a-syn), a small 14-kDa protein, encoded by the SNCA gene, which is the primary constituent of Lewy body deposits present in the brain of subjects affected by PD [6]; (ii) ubiquitinated protein inclusions of a transactive response DNA-binding protein of 43 kDa observed in the motor neurons of individuals with ALS [7,8]; (iii) Aβ plaques, consisting of 39- to 42-aa-long Aβ peptide fragments and neurofibrillary tangles, formed by tau oligomers which, in Alzheimer’s disease (AD), are shown to be present outside and inside neurons, respectively [9,10]; (iv) mutant huntingtin protein aggregates reported as one of the main features of Huntington’s disease (HD) and present in the cytoplasm and in the nucleus of the nervous cells throughout the brain [11]; (v) prion protein PrPSc accumulation, a 253-amino acid protein encoded by the gene for PrP (PRNP) located in chromosome 20, which also shows a synaptic pattern of deposition and is detected in the so-called “prion diseases” [12]; (vi) FET proteins, a protein family which includes FUS, the fused-in sarcoma protein, EWSR, the Ewing sarcoma RNA-binding protein 1, and TAF15, the TATA-binding protein-associated factor 15 (TAF15) [13], reported to form cytoplasmic aggregates found in neurological diseases, such as frontotemporal lobar degeneration (FTLD) and ALS [14,15].

Unfortunately, no successful treatment for NDDs has been developed so far. Due to the complex nature of NDD pathophysiology, a multimodal therapeutic approach may be needed. This aims to replace lost nervous cells, remove toxic deposits, and guarantee a safe environment essential for the survival and plasticity of these cells. In recent years, cellular therapy has been considered a novel potential therapeutic strategy for the treatment of NDDs [16,17,18].

Different cell types, such as embryonic stem cells (ESCs), adult stem cells, and induced pluripotent stem (iPS) cells have been tested for their ability to replace damaged cells and to restore tissue function after transplantation [19]. In the last few years, perinatal tissue-derived mesenchymal stem cells (MSCs), including placenta-derived, chorion, and umbilical cord-derived MSCs, have raised great interest due to the large accessibility, their ability to differentiate in several cell lineages (Figure 1), the absence of tumorigenicity after transplantation and, importantly, the lack of ethical problems initially limiting the availability of ESCs [20,21]. The isolation of stem cells from fetal material does not raise ethical concerns since these tissues are considered medical waste immediately after delivery and are readily accessible. Research material can be collected from tissues obtained from invasive diagnostic and treatment procedures throughout the pregnancy, planned terminations, and after the full-term vaginal delivery or cesarean section.

Studies focusing on perinatal stem cells are conducted worldwide. Several countries, such as the U.S. and China, have established their regulations on regenerative medicine products. Each European Union member state currently has specific regulations for research on embryonic stem cells and perinatal tissues. Perinatal stem cell clinical trials must be approved by the local bioethical commissions and be conducted in line with the EU clinical trials registration law [21]. Until April 2023, 1120 registered clinical trials had been using MSC therapies worldwide; however, only 12 MSC therapies have been approved by regulatory agencies for commercialization. Nine of the twelve MSC-approved products are from Asia, and the Republic of Korea is the country with the most approved therapies. Among them, there is NeuroNata-R, an autologous bone marrow MSC therapy with neuroprotective effects that relieves the progression of ALS [22].

The results obtained in preclinical studies using animal models, and even some clinical trials support the ability of MSCs to differentiate into neuronal lineage cells and to promote the functional recovery of injured neuronal tissue through different mechanisms including immunomodulation, trophic actions, neuroprotection, and the stimulation of angiogenesis [22,23,24,25,26]. However, naturally, MSCs do not spontaneously differentiate into a specific neural lineage cell type, but their differentiation into neurons can be promoted by combining multiple nerve-inducing factors. This is considered a generally expensive and complicated approach; moreover, most differentiated neurons in vitro are neuron-like cells without neural function [27,28,29,30]. Therefore, the development of new strategies to differentiate MSCs into functional neuronal cells remains a major challenge in neuroscience. Moreover, the successful clinical application of MSC therapy to replace damaged neurons in NDD is still limited by multiple factors. They include the risk of host immune response, the safety profile, technical problems associated with mass production, high manufacturing cost, and contamination, as well as the need to develop manufacturing processes ensuring the high therapeutic quality and purity of cells in line with the current good manufacturing practice (GMP) standards. This review summarizes the recent advances in biological and translational research regarding the potential use of perinatal tissue-derived MSCs in the treatment of NDD, as well as their clinical limitations.

## 2. Cell Therapy in Alzheimer’s Disease

AD is the most prevalent cause of dementia, accounting for 50–70% of dementia cases worldwide [31]. The disease is characterized by the deterioration of cognition, function, and behavior, with an insidious onset, usually beginning with the loss of memory of recent events, and a slow progression [31,32]. Most patients are affected by the sporadic form of the disease (>95) for which several genetic risk factors have been identified, whereas a small number of subjects present inherited mutations affecting the processing pathways of the amyloid-β (Aβ) peptide [33]. Two main pathological features characterize AD. One is the extracellular Aβ plaques (also known as “senile plaques”), composed of Aβ peptides of 39 to 42 aminoacidic residues [34,35]. The other one is the intracellular neurofibrillary tangles, composed of hyperphosphorylated tau protein, which assemble to form the so-called “paired helical filaments” [36]. The consequence of these biological alterations is neurodegeneration leading to synaptic and neuronal loss and ultimately to macroscopic atrophy. Other pathological processes including astrogliosis, microglial activation, and cerebral amyloid angiopathy are frequently found and contribute to the disease onset and progression [37,38,39,40].

The risk of AD is estimated to double every 5 years after the age of 65 [31] and the number of subjects affected by AD is predicted to triple by 2050, reaching millions of patients by 2050 [41], thus significantly contributing to the increase in the risk of disability. Both the socioeconomic and family burden of AD have prompted the World Health Organization to declare AD as a global health priority [42].

Despite the dramatic worldwide impact of AD, there are still no therapeutic approaches able to effectively reverse or counteract the progression of the disease. So far, only acetylcholinesterase inhibitors such as donepezil, galantamine, rivastigmine, and the N-methyl-Daspartate receptor antagonist memantine are available in the US and EU. Aducanumab is a human IgG monoclonal antibody targeting Aβ fibrils and soluble oligomers aimed at clearing the Aβ plaques and contributing to slow AD development [43,44]. This drug, considered the first disease-modifying treatment for AD, has been approved under the fast-track pathway by the US Food and Drug Administration. On the contrary, the European Medicines Agency recently recommended the refusal of its marketing authorization due to the conflicting results from the two phase III clinical trials that have concluded (EMERGE and ENGAGE) [45,46,47]. Indeed, the Committee for Medicinal Products for Human Use stated that the risk–benefit balance was unfavorable [48]. The company withdrew the application on 20 April 2022 [48]. In this scenario, the development of novel therapeutical approaches emerges as an urgent need.

Several pieces of experimental evidence support the idea that the transplantation of MSCs is associated with an improvement in synaptic plasticity [49,50,51,52,53,54,55]. MSCs have also been reported to ameliorate cognitive performance in different animal models of NDDs, including AD [56,57,58,59,60,61,62], thus suggesting that this can represent a promising therapeutic strategy. The standard source of human MSCs preferentially used for clinical applications is represented by bone marrow (BM). Although BM cell harvesting is considered a safe procedure, it is invasive and requires the need for general anesthesia, with related potential complications and risks [63]. Moreover, other complications may be present including the frequent presence of acute and/or chronic pain at the sites of aspiration, anemia, vasovagal reaction, and infection [64]. To try to overcome these challenges, a device for a rapid, poorly invasive harvest of BM for both autologous and allogeneic use, called “MarrowMiner”, has been developed and recently approved by the US Food and Drug Administration [65].

On the other hand, gestational tissue-derived human MSCs, which (i) can be easily obtained in large amounts; (ii) do not need crude procedures; (iii) and do not raise ethical concerns have been increasingly considered a valid and suitable source of MSCs for clinical applications. Among gestational tissues, human umbilical cord (hUC) is considered one of the best sources of MSCs for the reasons reported above, but also for their low immunogenicity and high proliferative capacity associated with a very low potential to generate tumors [66,67]. Thus, hUC-MSCs have gained much attention and interest for their use in regenerative medicine [68] and as an attractive novel therapeutic approach in different NDDs, including AD [69,70].

### 2.1. Key Findings from Preclinical Studies

Several preclinical studies have been performed using different animal models of AD, in which hUC-MSCs have been either transplanted or injected to counteract Aβ plaque formation, reduce inflammation, and/or ameliorate cognitive performance. As shown in Table 1, almost all the studies carried out in the last ten years have reported that the treatment with hUC-MSCs resulted in a reduction in Aβ deposition. Moreover, in 2013, Yang and coworkers transplanted (by direct injection) a suspension of hUC-MSCs, previously differentiated in vitro into neuron-like cells, into the hippocampus of heterozygous AβPPswe/PS1dE9 double-transgenic mice with C57BL/6 background, an animal model widely used in the study of AD [71]. By using thioflavin S staining to detect Aβ deposition, they found that in mice treated with the differentiated hUC-MSCs (one injection), Aβ deposition was significantly reduced in both the hippocampus and cortex compared with the animals injected with the vehicle. This effect was associated with M2-like microglial activation; thus, a stem cell modulation of neuroinflammation could be hypothesized. Moreover, the differentiated hUC-MSCs also improved the performance of mice in the Morris water maze test without increasing the swimming speed, indicating an effect on the cognitive processes [71]. In the same AD animal model, following a single administration into the hippocampus, hUC-MSCs survived up to eight weeks [72]. On the contrary, following three cell injections into the cisterna magna every four weeks, hUC-MSCs were still present in the brain parenchyma 8 weeks after the last treatment. These findings correlated with the beneficial effects. Indeed, the repeated hUC-MSC administration was more effective in reducing the levels of soluble and insoluble Aβ, analyzed by immunoblotting, compared to the single treatment [72]. Moreover, both the single and the repeated administrations of hUC-MSCs almost doubled the population of adult hippocampal neural stem cells (NSCs) (1.55- and 2.06-fold, respectively). Similar results were obtained for the number of mature neurons in the dentate gyrus. The suppression of growth differentiation factor (GDF)-15 in hUC-MSCs by using small interfering RNA reduced the in vitro proliferation rate of neural stem cells (NSCs), thus underlining the role of paracrine factors in the stem cell-mediated beneficial impact on neurogenesis and synaptic activity [72].

To better clarify the paracrine mechanisms underlying the therapeutic role of hUC-MSCs, another study has been carried out using SAMP8 mice, a senescence-accelerated mouse model of AD [70]. It demonstrated that hepatocyte growth factor (HGF), secreted by the transplanted MSCs, was able to regulate the expression of key AD-related proteins including tau hyperphosphorylation through the activation of the cMet-AKT-GSK3β signaling pathway [70]. HGF was also involved in the hUC-MSCs-mediated improvement in SAMP8 mice cognitive functions evaluated by a series of different behavioral tests such as the Morris water maze, Y-maze, open field and object recognition, and the shuttle box, thus supporting the key role played by the “secretome” of these cells [70].

Recently, autophagy, a highly regulated process involved in both physiological and pathological conditions, has been claimed to play a significant role in some key events regulating the fate of MSCs, such as cell differentiation, stemness maintenance, and senescence [73,74]. On the other hand, MSCs themselves can modulate autophagy in those cells involved in the AD pathophysiological events [73]. Interestingly, the two mechanisms have been recently reported to participate in the beneficial effects caused by hUC-MSCs in AD animal models [75]. Human umbilical MSCs, previously differentiated in vitro into neuron-like cells were transplanted, through injection into the left lateral ventricles, in APP/PS1 transgenic mice. The treatment improved the learning and memory ability of the animals, evaluated by the T-maze test, and reduced both Aβ production and cell death in the transgenic mouse brain [75]. When autophagy was inhibited by knocking down Beclin 1 expression, an essential player in the autophagic process [76], in the transplanted hUC-MSCs, these neuroprotective effects were lost [75]. Furthermore, 14 days post-transplantation, the transplanted hUC-MSCs reached the damaged hippocampus, and they expressed markers of differentiated neurons. These features were again blocked by the autophagy inhibition [76]. Xu et al. also confirmed the role of autophagy in the hUC-MSC-mediated neuroprotective effects [77]. In this in vitro study, the conditioned medium (CM) of these gestational tissue-derived stem cells modulated the autophagy of BV2 microglial cells stimulated by Aβ25–35 by reversing the increase in the LC3II/I ratio and rescuing the inhibition of Beclin-1 and p62 induced by the toxic oligomers [77]. In addition, the cell death of SH-SY5Y cells cultured with the medium from BV2 cells treated by hUC-MSC-CM plus Aβ25–35 was significantly lower compared to the control medium [77].

Another biological process that is emerging as a potential pathophysiological event in the development of AD is cellular senescence reported to mediate Aβ- and tauopathy-induced neurodegeneration [78,79]. Cellular senescence can also affect the potential therapeutic efficacy of transplanted MSCs; indeed, irrespective of their source, they inevitably acquire a senescent phenotype after serial in vitro passages [80]; thus, research is needed to counteract this phenomenon to increase their clinical performance. Zhang and coworkers, for the first time, induced the overexpression of Forkhead box Q1 (FOXQ1), a protein highly expressed in different cell tumors and down-regulated in oncogene-induced senescence [81], in hUC-MSCs (FOXQ1- hUC-MSCs). They showed that in FOXQ1- hUC-MSCs, the expression of positive senescence-associated genes, such as p16, p21, and p53 (*p* < 0.05) was down-regulated compared to the controls. In contrast, the expression of the silent information regulator 2, homolog 1 (SIRT1), and proliferating cell nuclear antigen (PCNA), known to be negatively associated with senescence, was up-regulated [81]. When FOXQ1- hUC-MSCs were transplanted (by injection through the tail vein) in AD mice, the animals showed a better performance in the spatial learning–memory test carried out in the Morris water maze compared to the control mice. No differences were recorded in the swimming speed, indicating, again, an effect on the cognitive processes. FOXQ1 overexpression also enhanced the migration and survival of hUC-MSCs in vivo. The evaluation of tumor development in AD mice showed that FOXQ1- hUC-MSCs was a safe treatment [81]. Interestingly, similar results on learning and memory were obtained in transgenic AD mice treated with a combination of resveratrol, a sirtuin 1 (SIRT1) activator, and hUC-MSCs [82]. The combined treatment enhanced the neurotrophin expression levels [Brain-Derived Neurotrophic Factor (BDNF), Nerve growth factor (NGF), and Neurotrophin-3 (NT-3)] and neurogenesis in the hippocampus of AD mice. Furthermore, resveratrol plus hUC-MSCs up-regulated the hippocampal expression of SIRT1 and PCNA, whereas it down-regulated that of p53, p21, and p16. All these effects were higher compared to resveratrol or hUC-MSCs given alone [82].

Beyond hUC-MSCs, MSCs deriving from another gestational tissue, the placenta, are gaining increasing interest in regenerative and reparative medicine. In 2013, Yun and coworkers transplanted an Aβ1–42-infused mouse model with MSCs isolated from chorioamniotic membrane (PD-MSCs) (1 × 10^5^, 5 × 10^5^, and 1 × 10^6^ cells per mouse) via intravenous injection into the animal tail vein. They found that PD-MSCs significantly improved the performance of the Aβ1–42-infused mice on the water maze test, the probe test, and the passive avoidance tests at all the cell concentrations used [83]. Interestingly, these beneficial effects on learning and memory were present over a prolonged period. Moreover, PD-MSCs also constrained the amyloidogenic process by (i) the reduction in the expression of the beta-site amyloid precursor protein cleaving enzyme (BACE), involved in the formation of the Aβ42 peptide found within amyloid plaques [84]; (ii) the reduction in the expression of β-amyloid precursor protein (APP); and (iii) the decrease in the activity of γ-secretase. These cells also exerted an anti-inflammatory effect by inhibiting cytokine secretion such as interleukin (IL)-1β, IL-17, tumor necrosis factor (TNF)-α, and IP-100, which was increased in Aβ1–42-infused mice and which are known to play a significant pathological role in AD [83]. These results were in agreement with the findings from more recent studies carried out by using transgenic mouse models of AD infused with human placenta amniotic membrane-derived MSCs (hAMMSCs) [85,86]. Human AMMSCs were able to significantly ameliorate the cognitive functions of the transgenic mice evaluated by the Morris water maze [85,86] and Y-maze tests [86]. Cell transplantation also dramatically reduced the Aβ deposition and plaque load present in the cortex and hippocampus of the transgenic mice [85,86]. A protective effect against oxidative stress was also pointed out. Indeed, hAMMSC transplantation caused an increase in the reduced glutathione/oxidized glutathione ratio (GSH/GSSG) present in the transgenic mice compared to the control animals [85], thus contributing to the maintenance of the homeostasis of the brain redox status. The infused cells also enhanced the activity of superoxide dismutase (SOD) [85], a key antioxidant enzyme, thus reducing the formation of malondialdehyde, an end product of lipid peroxidation with a recognized neuronal toxicity [87,88], thus indicating that hAMMSCs could ameliorate learning and memory in AD also by modulating the oxidative stress.

Finally, hAM-MSCs derived from human term placenta have been shown to exert more effective immunomodulatory and paracrine effects than BM-MSCs [89]. In a transgenic mouse model of AD, APPswe (Tg2576) mice, Kim and coworkers reported that the transplantation of hAMMSCs triggered the recruitment of microglial cells in the early phase after the cell infusion. This effect was followed by an hAMMSCs-mediated control on resident microglial cells, which, after the initial phase, were lower despite the development of a proinflammatory milieu [90], thus supporting an immunosuppressive role for these stem cells. Preclinical studies on the transplantation of gestational tissue-derived MSCs in AD are summarized in Table 1.

**Table 1 ijms-25-00976-t001:** Key findings from preclinical studies with transplantation of perinatal tissue-derived MSCs in Alzheimer’s disease murine models.

MSC Type	Cell Number	Passage Number (P)	Mouse Model	Route of Transplantation	Key Findings	Ref.
neuron-like cells derived from hUC-MSCc	~5 × 10^4^	P2-P6	APP/PS1 mouse	Single Injection into the bilateral hippocampus	Cognitive decline (↑); Synapsin I level (+); IL-4 expression (+); M2-like microglial activation and neuroprotection; Proinflammatory cytokines expression (such as TNF-α and IL-1β, TNF-α, IL-6, and IL-1β) (−); Aβ clearance by invoking Aβ degrading factors (+)	[71]
WJ-MSCs	~2 × 10^6^	-	APP/PS1 mouse	IV	Cognitive, learning, and memory functions (↑); Aβ deposition in cortex and hippocampus (−); Aβ 40 and Aβ 42 levels (−); Brain expression of IL-10 (+), IL-1β and TNFα (−); Number of resident-activated microglial cells after initial stage (−)	[91]
hUC-MSCs	~2 × 10^6^	-	Tg2576 mouse	IV	Cognitive function (↑); Oxidative stress in the hippocampus (−); Cell proliferation, newborn cell survival, and neurogenesis in the hippocampus (+); Expression of Sirt1, BDNF, and SYN and neuroprotection (+)	[92]
hUC-MSCs	~1 × 10^6^	P4	Tg2576 mouse	IV	Enhanced engraftment of hUC-MSCs in the hippocampus (+); Cognitive function (↑); Neural apoptosis (−) and neurogenesis (+) in the hippocampus; Levels of BDNF, NGF, and NT-3 (+); Levels of SIRT1, PCNA (+); levels of p53, ac-p53, p21, p16 (−)	[82]
hUC-MSCs infected with negative control lentivirus (huMSCs-shNC) or lentivirus expressing shRNA inhibiting the gene Beclin-1 (huMSCs-shBecn 1)	~1 × 10^6^	P3	APP/PS1 mouse	Injection into the left lateral ventricles	Autophagy of hUC-MSCs (−) associated with no amelioration of impaired learning and memory; no reduction in expression of APP and PS1; drop in migration and antiapoptotic functions; no improvement in synaptic transmission	[75]
hUC-MSCs overexpressing FOXQ1	~8 × 10^5^	P3 and P15	APPV717I mouse	IV	Spatial learning–memory (↑); Oxidative effects (↑); Efficacy of hUC-MSC transplantation (+)	[81]
WJ-MSCs	~2 × 10^5^	P5-P6	5XFAD mouse	Stereotactic injection into the left hippocampus	Proteasome activity (+); Accumulation of ubiquitin-conjugated proteins (−)	[93]
hUCB-MSCs	1 × 10^5^ cells	P6	5XFAD mouse	Intracerebroventricular injection	Spatial learning and memory deterioration (↑); Tau hyperphosphorylation through GAL-3 secretion (−)	[69]
hUCB-MSCs	5 × 10^4^	-	APP/PS1 mouse	Repeated intrathecal administration into the cisterna magna (up to 3 times)	Aβ levels in the brains (−); Adult NSC proliferation and differentiation (+); Neurogenesis and synaptic activity in hippocampal neurons (+)	[72]
hUCB-MSCs	1 × 10^6^	P6	5XFAD mouse	Intracerebroventricular injection	SYP and PSD-95 (+); Aβ42-induced synaptic dysfunction regulating TSP-1 release (−)	[94]
WJ-MSCs and hUCB-MSCs cocultured with SVZ-derived NSCs	1 × 10^6^	-	5XFAD mouse	-	Expression of activin A and GDF-15 (+)	[95]
PD-MSCs	1 × 10^5^, 5 × 10^5^, 1 × 10^6^	-	A*β*1–42-infused mouse	Intracerebroventricular injection	Memory and learning impairment (↑); Antiamyloidogenic effect; Neuronal cell death (−); Neural differentiation (+)	[83]
hAM-MSC	1 × 10^5^	-	APP/PS1 mouse	Stereotactic injection into bilateral hippocampi	Spatial learning and memory impairments (↑); Aβ deposition (−); Stimulation of microglia activation (+); Modulation of neuroinflammation: levels of IL-1β and TNF-α (−), levels of IL-10 and TGF-β (+); Aβ-degrading enzyme expression (NEP, IDE, and MMP9) (+); Neurogenesis and synaptic plasticity mediated by BDNF (+)	[96]
hAM-MSC	~1 × 10^6^	P4	C57BL/6J-APP mouse	I.V.	Spatial learning and memory function impairments (↑); deposition of Aβ (−); Oxidative stress (−): GSH/GSSG ratio (+), SOD activity (+), and brain MDA level (−)	[85]
hAESCs	~2 × 10^6^	P4	Tg2576 mouse	Bilateral intracerebral administration	Cognitive deficits (↑); Number of amyloid plaques (−); BACE activity (−)	[90]
hAM-MSC	-	P3	Tg2576 mouse	I.V.	Activation of microglial induced by Aβ25–35 (+); Memory function (↑); Number of Aβ plaques (−); Number of resident and activated microglia (+); IDE and MMP-9 secretion	[86]

Human umbilical cord mesenchymal stem cells (hUC-MSCs); Wharton’s Jelly Mesenchymal Stem Cell (WJ-MSCs); subventricular zone (SVZ); neural stem cells (NCs); placenta-derived mesenchymal stem cells (PD-MSCs); human umbilical cord blood-derived mesenchymal stem cells (hUCB-MSCs); human amniotic epithelial stem cells (hAESCs); human amniotic membrane mesenchymal stem cells (hAM-MSC); (↑) improvement of disease signs; (+) increase; (−) reduction.

### 2.2. Cell Therapy in Alzheimer’s Disease: Key Findings from Clinical Trials

The role of gestational tissue-derived hMSCs has been also evaluated by clinical trials, even though only a few have been completed. A phase 1 clinical trial was conducted in nine patients with mild-to-moderate AD in which hUCB-MSCs were stereotactically administrated in the brain. The participants were divided into two groups: low- and high-dose groups who received, in a single administration, 3.0 × 10^6^ cells/60 μL and 6.0 × 10^6^ cells/60 μL, respectively. The study demonstrated that hUCB-MSC injections into the hippocampus and precuneus by stereotactic surgery were safe and well tolerated. However, they were not able to evoke the beneficial effects related to the AD pathophysiological process seen in animal studies during the two-year follow-up. Kim and coworkers tried to explain the disparity between the animal and human studies considering the AD microenvironment differences and sensitivity differences between the detection methods of soluble amyloid or diffuse amyloid plaques. Moreover, the lack of a control group and the small number of participants represent some of the limitations of this study (NCT01297218; NCT01696591) [97]. The safety, dose-limiting toxicity, and exploratory efficacy of repeated intracerebroventricular injections of NEUROSTEM^®^ (human umbilical cord blood-derived mesenchymal stem cells) into the lateral ventricle via an Ommaya reservoir versus placebo have been investigated by a combined phase 1/2a clinical trial. Three repeated injections were performed at 4-week intervals in subjects with AD, who were followed up to 12 weeks after the first hUCB-MSC injection and an additional 36 months in the extended observation study (ClinicalTrial.gov; Identifier: NCT02054208; NCT03172117). The study consists of two stages: stage 1 involved dose-escalation in 9 subjects (3 subjects for the low dose and 6 subjects for the high dose) while stage 2, randomized and multiple-dose cohort parallel-designed, involved 36 subjects (24 subjects for the high dose and 12 subjects for the placebo). The observation of reduced levels of total tau, phosphorylated tau, and Aβ42 in the cerebrospinal fluid (CSF) samples collected 1 day after hUCB-MSC injections and their increase 4 weeks following hUCB-MSC transplantation suggested that repeated administrations are necessary to maintain the therapeutic effects exerted by the hUCB-MSCs. A limitation of this study is that the hUCB-MSC injection dose used in this trial was based on mouse studies, by referring to the mouse-to-human CSF volume ratio [98]. Detailed studies to evaluate the optimal cell dose in AD patients should be performed.

Some clinical investigations are still actually ongoing. For example, a phase I, prospective, open-label trial wants to evaluate the safety, tolerability, and exploratory outcomes of umbilical cord-derived, allogeneic hMSC infusions. The study involves a total of four doses, intravenously administered once about every 13 weeks within a year, of approximately 100 million cells per infusion (NCT04040348).

Evaluating the safety and the potential therapeutic effects of CB-AC-02 intravenous transplantation in patients with AD is the purpose of a randomized, double-blind, placebo-controlled, phase I/IIa clinical trial, not yet completed (NCT02899091). In this study, 24 participants are divided into two groups: the treatment group receives 1 or 2 (after 4 weeks) I.V. infusions of 2.0 × 10^8^ P-MSCs and is compared to the placebo group.

A summary of the clinical trials registered on ClinicalTrials.gov evaluating the transplantation of gestational tissue-derived MSCs as an AD treatment can be found in Table 2.

Despite all favorable findings found in MSC-AD therapy, this treatment has several limitations. The translation of the results of basic research into clinical investigations is problematic considering the inability of animal models to completely reproduce the entire pathophysiology of the disease. The anatomy and microenvironment of the brain exhibit significant differences among animal models and AD patients, thus, the precise characterization of the beneficial effects of stem cells in human AD has proven difficult. Furthermore, most preclinical studies use transgenic mice expressing familial AD (fAD) mutations, while clinical studies usually involve patients with sporadic late-onset forms of AD (sAD) [99]. This limits the knowledge of how stem cell therapy would act in a patient-specific manner. Thus, the development of a successful AD model, notably for sAD, may be crucial to reduce the frequency of unsuccessful clinical trials and allow for a more efficient transition from basic research to clinical trials. Individualization is another key factor that hampers pre-clinical studies from being applied in clinical settings. For cell and animal models, there is less of a concern for the variation in each individual. The employment of hUCB-MSCs in clinical studies is the most recurrent even if their use (the number of injections and delivery routes) must be optimized. Similarly, further research is required to prove the safety and efficacy on a long-term basis [100]. At the time of AD clinical diagnosis, the neuronal loss and pathological proteins have already accumulated in many brain regions; thus, it is difficult to reverse the disease process. Furthermore, in many clinical trials, patients receive only stem cell infusions several times, whereas they might need multiple infusions over an extended period. In many trials, AD patients are usually advanced in age and autologous MSCs may suffer from senescence, which reduces their regeneration capability. The intravenous route is the most used in clinical trials with AD patients and a large amount of intravenously administered stem cells will become detained in the lung and the spleen [101]; thus, only a limited number of stem cells can reach the brain. Moreover, the hostile microenvironment also hinders the survival of infused stem cells. Altogether, current MSC clinical trials have important limitations that require immediate renovation in the field.

There is, so far, no other therapeutic approach able to provide the pleiotropic effects of stem cells, but there is an urgent need to maximize the effects of stem cells to replace the missing nerve cells. Shortly, this may occur through the development of new stem cell-based technologies (including the use of engineered stem cells, biological scaffolds, and other new materials [102], such as nano-sized microvesicles), which will drastically change this field. At the same time, to improve their results, AD clinical trials should be more carefully designed, by targeting different patient populations, and in the early phase or prodromal phase of the disease.

More research is also needed to promote accelerated approval for the validation of stem cell therapy for AD and to better understand its mechanisms of action. The Food and Drug Administration (FDA) has not yet approved stem cell therapy as a treatment for AD and the cost of this therapy may represent a barrier for many patients.

Although several clinical trials have been conducted or are ongoing to evaluate the efficacy along with the safety profile of MSC therapy in NDD, few published results are available [103]. In the NCT01297218 and NCT01696591 studies [97], the administration of allogeneic hUCB-MSCs to the hippocampus and precuneus by stereotactic injection in patients with AD had no dose-limiting toxicity during 2 years of follow-up. The common unwanted events were wound pain from the surgical procedure, headache, dizziness, and postoperative delirium, thus supporting the safety and feasibility of MSC transplantation [97]. A phase I clinical trial by Kim et al. in nine patients with mild-to-moderate AD also showed that the intracerebroventricular transplantation of human UCB-MSCs could be safe and feasible [98]. The common unwanted events were fever, headache, nausea, and vomiting alleviated within 36 h [98]. Altogether, several clinical trials have been performed addressing the safety of MSC transplantation in patients with NDD. Overall, the results of these studies support the safety and feasibility of the transplantation of both autologous and allogeneic MSCs [103]. Published reports have shown that no tumors were formed because of the transplant, and no deaths occurred because of the therapeutic intervention. However, despite promising results regarding the efficacy in various clinical trials and their proven safety and feasibility, many questions remain open. The cell type, administration route, dose, and frequency are very different between the individual studies. Even among studies for the same disease that use the same cell type, there is large variability between these parameters; thus, all results should be read with caution.

## 3. Cell Therapy in Huntington’s Disease

Huntington’s disease (HD) is a progressive neurodegenerative disorder. It is an autosomal dominant genetic disorder that shows pathological and clinical features. Both neuronal malfunction and cell death probably cause progressive motor disability, which includes chorea, bradykinesia, incoordination and rigidity, cognitive impairment, behavioral change, and psychiatric disorders. In particular, the death of GABAergic medium spiny neurons (MSNs) is involved in the onset of HD manifestation [104,105].

Huntington’s disease is caused by a CAG triplet repeat expansion in the Huntington gene, which encodes an expanded huntingtin (HTT) protein. This protein is essential for embryonic development, and it is involved in cellular activities, such as vesicular transport and recycling, endocytosis, endosomal trafficking, autophagy, and transcription regulation; however, its normal function(s) is still incompletely understood. In the general population, the CAG repeat length is of 16–20; 40 or more CAG repeats are pathogenic. Individuals with an expanded HTT allele can become symptomatic at any time point in their life. In the vast majority of cases, the clinical course of HD slowly begins in adulthood, typically in the mid-40s. The disease is inherited in an autosomal dominant manner with age-dependent penetrance and individuals at risk can be identified before clinical onset by predictive genetic testing. Longer CAG repeats predict an earlier onset, while the length of the CAG repeat seems to contribute less to the rate of progression [106].

The pathological signature of HD is the formation of intranuclear inclusion bodies, which are large aggregates of abnormal HTT in neuronal nuclei, cytoplasm, dendrites, and axon terminals [107]. Mutant HTT causes the decreased transport and release of corticostriatal BDNF. An increased stimulation of extrasynaptic glutamate receptors occurs and the reuptake of glutamate by glia is reduced. As a result, excitotoxicity and enhanced susceptibility to metabolic toxic effects take place. Finally, activated microglia produce increased inflammatory activity [108]. Epidemiologically, HD shows a prevalence of 5–7 per 100,000 in most White populations. It is a fatal disease; death generally occurs 15–20 years from its onset [105], and there is no effective cure yet. At present, most drugs are just able to ameliorate the symptoms and they are not able to stop the neurodegeneration [109].

For these reasons, stem-cell-based regenerative medicine is gaining increasing interest in HD treatment. Cell-based therapies aim to restore brain functions by replacing lost/dying neurons as well as giving neurotrophic aid to diseased tissue.

Several studies suggest the potential efficacy of MSCs in the treatment of HD [110,111,112,113,114]. A recent meta-analysis confirmed the positive effect of MSCs on HD animal models overall, as reflected in morphological changes, motor coordination, muscle strength, neuromuscular electromyography activity, cortex-related motor function, and striatum-related motor function, while cognition was not changed by MSC therapy [115]. Among different MSC sources, bone marrow MSCs were the most investigated cells and were effective in improving motor coordination. However, while capable of reducing behavioral and histological deficits in the R6/2 mouse model of HD, bone marrow-derived MSCs did not generate new neurons following transplantation in the mouse striata. For this reason, stem cells from other sources, mainly from gestational stem cells, are gaining interest. In 2001, an interesting study [116] showed an increase in the survival time of B6CBA-TgN(Hdexon1)62Gpb mice, transgenic mice displaying a phenotype that mimics many of the features of HD in humans, by using the so-called “megadoses” of human umbilical cord blood cells (71–74 × 10^6^ and 100–105 × 10^6^). The umbilical cord is an attractive source of MSCs, as they can be harvested at a low cost and they have a higher harvest rate when compared to cells from bone marrow; thus, it is possible to isolate a great number of cells. This limits the time and number of passages in culture to produce clinically relevant numbers of cells for transplantation [117]. Umbilical cord stem cells have an advantage over other types of adult stem cells as they do not require human leukocyte antigen (HLA) matching. Further, cord blood is easily cryopreserved, and this is useful for bio-banking and expansion [117].

Glial cell line-derived neurotrophic factor (GDNF) and BDNF are principal mediators involved in HD neuropathology [118]. GDNF has also been shown to provide striatal neurons with neuroprotective support against excitotoxic lesioning in the quinolinic acid (QA) lesion rodent model of HD. However, GDNF has demonstrated only the partial protection of the GABAergic striatal neurons from QA-induced cell death. These results probably reflect GDNF difficulties in achieving efficient delivery to striatal neurons in vivo, thus limiting the potential of this factor in counteracting the neurodegenerative process of HD [119]. McBride and colleagues [120], using the alternative 3-nitropropionic acid (3-NP) animal model of HD, demonstrated that the delivery of GDNF to the striatum using a recombinant adeno-associated viral (AAV) vector could protect striatal neurons from degeneration. This suggests that the neuroprotective actions of GDNF on the striatal neurons involve not only the protection against excitotoxicity but may be also effective against mitochondrial inhibition. It has been reported that GDNF may provide striatal neuroprotection by promoting the expression of various antiapoptotic factors, including X-linked inhibitors of apoptosis, bcl-2, and bcl-xL [121,122].

BDNF is reduced in HD neural cell models, and genetic and pathogenetic mouse models of HD and in human post-mortem material [123], and the over-expression of BDNF has been verified to improve the HD phenotype in the mouse model [124]. A possible mechanistic link between BDNF with HD has been established. BDNF is essential for the corticostriatal pathway, and cortical BDNF is involved in the survival and differentiation of striatal neurons both at physiological and pathological levels. Moreover, the normal huntingtin protein contributes to the physiological control of BDNF synthesis and transport in the brain. Both these processes are disrupted in HD patients, who have huntingtin mutant protein. Several lines of evidence show that HD patients, together with being affected by the toxicity of mutant huntingtin, also are characterized by decreased normal huntingtin activity, which may reduce cortical BDNF gene transcription [125]. Altogether, all these lines of evidence indicate that HD involves profound changes in BDNF levels and that attempts to restore these levels are therapeutically interesting. Results of BDNF supplementation are promising but this approach raises several different problems, mainly related to the unstable nature of BDNF, which only in small amounts can cross the blood–brain barrier. In this context, some drugs that enhance BDNF production in the brain are being studied, as the production of BDNF mimetics. Finally, interesting new perspectives have been suggested by the findings that physical exercise and diet markedly increase endogenous BDNF levels in the hippocampus and cerebral cortex [125].

In HD patients, early studies have mostly shown that BDNF blood levels were decreased compared with controls [126,127,128]. Ciammola et al. further observed a significant association between the BDNF level and the clinical severity of the disease and cognitive performance [126]. However, these findings were not sustained by other studies. Zuccato and colleagues found no difference in the BDNF levels in serum and plasma in a large cohort of patients [129]. Other studies with plasma samples have confirmed this finding [130,131] and further detected no significant associations between the plasma BDNF and clinical signs of HD [132,133]. Overall, the current evidence does not support the value of BDNF as a robust biomarker of HD. Peripheral BDNF levels did not decrease as expected; this may be partially due to the region-specific secretion properties and its complex originations. It has been reported that a BDNF decrease mostly occurred in the striatum, but not in the cortex, thus suggesting that the cortex was unaffected, or even compensated for the deficit. While the BDNF level changes remain conflicting in blood, interestingly, the analysis of the DNA methylation level at BDNF Promoter IV in the whole blood revealed a significant association with anxiety and depression symptoms, but not with any motor or cognitive performances. This relationship between DNA methylation alterations of BDNF Promoter IV and neuropsychiatric symptoms deserves further validation [133].

In 2013, Fink et al. evaluated the efficacy of UC-MSCs in the R6/2 transgenic mouse model of HD [134] and they found that UC-MSCs (at a high passage) improved behavioral and neuropathological symptoms greater than low-passage cells. Interestingly, in these cells, they found that the reduced neuroprotection of low-passaged UC-MSCs compared to the high-passage cells was associated with a higher mRNA expression of BDNF in vitro, thus suggesting that the mechanism underlying MSC-mediated recovery is not only dependent on BDNF, but probably involves a host of other trophic and immunomodulatory factors. Accordingly, recent studies have demonstrated that bioactive factors produced by hAMSC in conditioned medium (CM-hAMSC) can provide protection in both in vitro and in vivo models of traumatic brain injury [135]. These factors enhance neuronal survival, decrease pro-inflammatory M1 and M2 microglia polarization, and induce neurotrophins involved in neuronal and vascular remodeling [135]. CM-hAMSC also contains lysine, taurine, alpha-aminoadipic-acid, and spermidine, which have well-documented neuroprotective effects [135], as well as anti-inflammatory molecules, such as IL-10 [136], and factors involved in the growth, differentiation, vascularization, and survival of neurons and synapses, such as transforming growth factor beta (TGF-β) [137], hepatocyte growth factor (HGF) [138], prostaglandin (PG)E_2_ [139], angiogenin [140], and leptin [141]. Giampà et al. (2018) [142] explored the potential immunomodulation of mesenchymal stem/stromal cells from the amniotic membrane (hAMSC) in treating high-grade dementia. They found that the peripheral administration of CM-hAMSC improved neuropathological changes, behavioral impairment, and motor performance in the R6/2 mouse model of HD. The positive effects were not dependent on BDNF, and a reduced level of inducible nitric oxide synthase (iNOS) was observed in the striata of CM-hAMSC-treated mice. This suggests that a decrease in iNOS levels may delay disease progression in HD models.

Ebrahimi et al. (2018) [143] studied the efficacy of UC-MSCs against oxidative stress related to HD in vitro and in vivo. They found that these cells protect against 3-NP-lesioned rat models of HD, which can cause striatal atrophy and behavioral disturbances. They also detected the synthesis of GDNF and VEGF in UC-MSCs, with GDNF showing neuroprotection in HD models and VEGF showing neuro-survival properties in a dose-dependent manner. Preclinical studies on the transplantation of gestational tissue-derived MSCs in HD are summarized in Table 3.

The models most frequently used in preclinical studies are chemical and transgenic murine models. Among chemical models for inducing HD-like symptoms, QA or 3-NP are often used. QA can be found endogenously, where it binds and activates the N-methyl-D-aspartate receptor, a glutamate receptor and ion channel protein expressed in nerve cells. High concentrations of QA are neurotoxic, leading to neuronal cell death; thus, it is used to induce neurodegeneration in animal models, including HD. 3-NP induces neurotoxicity via oxidative stress in striatum neuronal mitochondria. The effects of these chemical agents are acute and variable and depend on the animal and may cause weight loss, lethargy, loss of motor control, and atrophy in the striatum associated with neurodegeneration and death. However, neither of these two chemical models reproduces the molecular events of HD [144]. Most widely transgenic mouse models used in preclinical studies include N1T1-82Q2, R6/2, and R6/2-J2, which have a short mutated amino-terminal fragment of human HTT. Both chemical and genetic models of HD have been used in MSC transplantation studies. Overall, these studies have reported behavioral and memory improvements, reduced brain damage and the amelioration of striatal degeneration, and the enhanced expression of several striatal growth factors. Most authors attribute these results to the neuroprotective effect of MSCs. However, several points still need to be considered using animal models. The design of animal studies and the characterization of transplanted cells are poorly standardized, thus complicating comparative analysis. Generally, many differences can characterize experimental protocols, including the HD animal model used, the origin of transplanted stem cells, the duration of in vitro stem cell expansion, the stem cell passages, cryopreservation methods, the number of cells for transplantation, the route of administration, the evaluation of stem cell migration and differentiation after transplantation [145]. So far, in study protocols for HD, only one cycle of cell transplantation is performed, and this is not compatible with neurodegeneration, which is a progressive process. Thus, stem cell-based therapies should be regularly performed. The starting point for therapy and the time intervals between cell transplantations can cause great variability in the results. In the future, the use of most standardized protocols may promote study comparison and reproducibility.

Overall, up to now, the positive effects of MSC therapy on HD animal models have emerged, as reflected in morphological changes, motor coordination, muscle strength, neuromuscular electromyography activity, cortex-related motor function, and striatum-related motor function, even if cognition was not significantly affected by MSC therapy [115]. However, data from animal studies did not completely clarify how transplanted cells regulate the expression of inflammatory cytokines, chemokines, and neurotrophic factors. These concerns should also be addressed before clinical trials. Finally, it is important to underline that HD therapy protocols using stem cells should be designed not only for treating the clinical onset of HD but also for disease prevention. In this context, the development of new methods for the assessment of mutated HTT in CSF may potentially serve as a biomarker for testing the efficacy of cell therapies for HD [146].

## 4. Cell Therapy in Parkinson’s Disease

Parkinson’s disease (PD) is caused by the loss of dopaminergic neurons in the substantia nigra of the brain. PD is more typically diagnosed in elderly patients and affects about 1–2% of the population over 70 years of age. Current pharmacological treatment includes the administration of dopamine precursors that improve the symptoms but cannot impede the disease progression. Recent research progress has suggested cell replacement therapy as a powerful innovative strategy to cure patients with PD. In recent years, it has made enormous progress in the field of regenerative medicine, and stem-cell-based therapy has been designed to replace lost dopaminergic neurons [147,148,149,150,151].

Concerning transplantation, dopamine replacing could follow two different protocols: (1) the in vitro pre-differentiation of stem cells toward dopaminergic neurons and (2) in vivo differentiation toward dopaminergic neurons after implantation. From the studies reported in the literature, the mechanisms of stem cell therapy on PD can be classified into two repair categories: (a) a direct repair pathway that includes the increase in endogenous neurogenesis through the differentiation of cells transplanted into neurons and/ or integration with neural circuits of the damaged brain [152,153,154]; (b) indirect repair through trophic factors as stem cells express several neurotrophic factors able to promote the neuronal differentiation of local stem cells. Several studies have shown functional improvements and neuroprotective and neurodegenerative effects after gestational stem cell transplantation to animal models of PD. In a rat model of PD [155,156,157], it has been shown that undifferentiated UC-MSCs prevented the degeneration of 48.4% of dopamine neurons and 56.9% of dopamine terminals from loss. Other investigators have described the functional benefit of differentiated hUC-MSC transplantation towards dopaminergic neurons able to alleviate motor symptoms [158,159,160,161]. Two interesting studies have used the combination of a particular cocktail, such as choroid plexus epithelial cell-conditioned medium, knockout serum replacement, and Lmx1a (a gene of homeodomain family members and neurturin (NTN) to facilitate the conversion of UC-MSCs to dopamine neurons. After cellular transplantation in animal models, such as hemiparkinsonian rhesus monkeys, improvements in behavioral deficits were detected [162]. In 2000, Kakishita et al. demonstrated the ability of human amniotic epithelial cells (hAECs) to synthesize in vitro catecholamines including dopamine and survive after implantation into the striatum of a rat PD model [163]. The same group, after a few years, investigated the neuroprotective effects of these cells, showing their ability to produce factors capable of promoting the survival of dopaminergic cells in vitro [164]. Several preclinical studies have been conducted to date to correlate the functional improvements and neuroprotective and neurodegenerative effects of the neurotrophic factors secreted by hAECs on Parkinsonian animal models. However, the mechanisms underlying the therapeutic benefits of hAECs which have been observed in vivo are yet to be elucidated.

The application of human amniotic fluid stem (hAFS) cells in PD was pioneered by Donaldson in 2009 [165]. It was reported that undifferentiated cells did not promote the development of fully differentiated dopaminergic neurons in culture or after transplantation into the PD rat brain. After a few years, different preclinical data showed that CD44+AFS cells induce the regeneration of dopaminergic neuron cell-like cells, increase migration distances, and improve animal behavior in a rat model of PD. These results reinforce the necessity of further studies to investigate the potential therapeutic effect of hAFS cells and support their use in clinical therapy [165]. A very interesting report has been published by Han Wool Kim et al. [166]. These authors investigated the therapeutic potentials of human placenta MSCs (hpMSCs) and their neural derivative human placenta-derived neural cells (hpNPCs) in a rat model of PD. The allogenic transplantation of hpNPCs was able to restore the physiological deficits of PD through dual mechanisms, i.e., neuroprotection and immune suppression. Surprisingly, in the last few years, the regulation of immune responses in PD is considered an important target in developing therapies, and hpMSCs could represent a very attractive source for cell therapies. More recently, another study has demonstrated the protective effects of BDNF-modified human umbilical cord mesenchymal stem cell-derived dopaminergic (DAergic)-like neurons in PD rats [167]. After cell transplantation, rats have ameliorated deficits in rotation through neuroprotection and anti-neuroinflammation mechanisms. These experiments have confirmed that BDNF supplementation may be an effective treatment for PD. The best results were found in groups that received both human Wharton’s jelly-derived mesenchymal stem cells (hWJ-MSCs) and levodopa-carbidopa, indicating that the transplantation of hWJ-MSCs with levodopa can be a good strategy for improving motor behaviors [168].

Taken together, these findings provide evidence that a number of challenges and problems regarding cell-based therapies must be addressed: (i) the long-term survival of stem cells in the host brain; (ii) the active integration of transplanted cells into a local neural network; (iii) a low risk of side effects; (iv) the physiological release of dopamine in the brain of patients. Preclinical studies with the transplantation of perinatal tissue-derived MSCs in PD murine models and clinical trials on perinatal tissue-derived MSCs in PD are summarized in Table 4 and Table 5, respectively.

## 5. Cell Therapy in Amyotrophic Lateral Sclerosis

Amyotrophic lateral sclerosis (ALS) is the most common motor neuron disease and is characterized by the loss of both upper and lower motor neurons. ALS has an incidence that varies between 1.2 and 4.0 per 100,000 individuals per year and it predominantly occurs in males [169]. The exposure of females and males to sex hormones has been shown to influence the disease risk or progression. Therefore, the correlation between genetics and sex has been widely investigated in ALS preclinical models and in large populations of ALS patients carrying the pathological expansion of a hexanucleotide repeat in chromosome 9 open reading frame 72 (C9orf72), which is the most common genetic mutation identified in familial ALS [170]. A recent meta-analysis on the sex differences in genetic mutations in ALS patients showed that in women, a higher prevalence of expanded C9orf72-related ALS occurs. This result may be explained by potential sex-related factors, such as environmental, lifestyle, or hormonal factors, that may moderate pathogenic mechanisms and determine an older age of onset and longer survival in women [171].

Current progress in regenerative medicine has proposed cell-based therapy as a novel treatment to cure ALS. Despite being in the early stages of clinical translation, numerous preclinical studies have investigated the neuroprotective mechanisms of stem cells in animal models. Most promising for this purpose are hUC-MSCs, which can survive readily after transplantation and have good migratory potential [172,173]. In 2008, Rizvanov et al. transplanted genetically modified hUC-MSCs in transgenic G93A mice adopted as an ALS animal model. The results obtained demonstrated that transplanted cells successfully grafted into nervous tissue and could differentiate into endothelial cells, forming new blood vessels. In this report, it was hypothesized that the neuroprotective effect could be derived from the delivery of various neurotrophic factors by newly formed blood vessels [174]. Similar data were confirmed by other groups that have focused their attention on the combined effect of the gene/ stem-cell approaches. In addition, multiple studies have labeled hUC-MSCs to detect transplanted cells in vitro and in vivo by magnetic resonance imaging (MRI) after intraspinal injection in a transgenic mouse model of ALS [175,176,177]. In line with these promising findings, over the years, several labeling strategies have been proposed. In this regard, it was described for the first time the paramagnetic labeling of hAFCs and their subsequent long-term tracking in a murine model of ALS. Surprisingly, the presence of double tracers has not altered the survival of hAFCs but has allowed for a correlation between in vivo and ex vivo data at different moments [178]. This study describes the therapeutic potential of hAFSCs in the treatment of ALS; however, additional preclinical trials are required to elucidate their benefits in clinical therapy for motor neuron disorders. Among the gestational stem cells, hAECs have also been proposed as an attractive source in cellular treatment for ALS. These stem cells are a heterogeneous population, containing several undifferentiated progenitor cells, which have not been extensively investigated. They constitute an ethically acceptable alternative to embryonic stem cells, with a comparable multipotentiality and a very low immunogenic response. The finding that hAFCs express and release numerous cytokines and neuro-glial factors [179] further promotes their application in the field of NDDs. A single preclinical study has reported the beneficial effects of their transplantation in terms of extended survival, the improvement of motor function, and decrease in neuroinflammation [180]. Recently, Wang et al. investigated the therapeutic potential of BDNF in ALS. They observed that motor neurons derived from hUC-MSCs overexpressing BDNF had a beneficial role in ALS model mice [181]. These results are interesting and show that the engineering of hUC-MSCs might offer great promise for new medical treatments. Preclinical studies with the transplantation of perinatal tissue-derived MSCs in ALS murine models and clinical trials on perinatal tissue-derived MSCs in ALS are summarized in Table 6 and Table 7, respectively.

Although several studies have shown that MSC transplantation results in disease improvement, the mechanisms by which the beneficial effects of MSC therapy arise are not entirely understood. Several mechanisms of repair and support, including cell replacement, trophic factor or gene delivery, and immunomodulation have been observed, sometimes in tandem [182]. Some studies have shown the ability of MSCs to differentiate into cells with neuron-like morphology, gene expression, and protein expression [183,184]. However, this phenomenon is still controversial, mainly due to the lack of evidence of functional synapse formation between trans-differentiated MSCs, and their therapeutic contribution is still uncertain. MSCs express or can be stably transduced to overexpress trophic factors which may promote endogenous restorative or regenerative processes, such as neurogenesis, gliogenesis, and synaptogenesis [185]. MSCs may play several immunoregulatory roles which may contribute to their beneficial effects in ALS. They reduce the proliferation of B cells, T cells, and natural killer cells, and impair the maturation of dendritic cells. Then, they can also affect immune cell function by reducing (i) antibody production by B cells, (ii) the activation of dendritic cells and T cells, and (iii) the secretion of natural killer cells [182]. In the central nervous system, MSCs migrate to areas of inflammation, reducing it. In experimental models of ALS, MSCs attenuate microglial activation and reduce astrogliosis [186,187,188].

ALS treatment currently focuses on medications that may slow the progression of the disease. Currently, three pharmaceutical compounds with an effect on ALS progression are approved in different countries: the glutamate antagonist Riluzole, the antioxidant Edaravone, and the recently introduced Sodium phenylbutyrate/Taurursodiol [189]. These drugs can prolong autonomy and increase survival by a few months. However, curative treatment options are still unknown. Based on the positive results of preclinical studies, over the last several years, MSC therapy has been proposed for ALS treatment. MSC infusion into ALS rodents ameliorates motor symptoms and improves the lifespan of animals. The intrathecal administration of MSCs in SOD1 mutant mice induced the proliferation of endogenous neural progenitor cells and modulated local inflammation. Clinical studies have shown that autologous or perinatal MSCs injected into ALS patients reduced disease progression in some individuals [190]. However, important hurdles should be overcome before the bench-to-bedside translation of these cellular therapies. Methodological challenges include adequate big trial cohorts, the heterogeneity of patient cohorts, and the lack of publication of previously performed stem-cell trials. Moreover, there is a distinct need for large multicenter and placebo-controlled trials. Studies combining MSCs with other therapies for NDDs are relatively rare, and if so, they are linked preferably with physical rather than pharmacological interventions. 

## 6. Functional Differentiation of MSCs towards Neuronal Lineage in Neurodegenerative Diseases: An Unmet Clinical Challenge

The potential efficacy of MSCs to restore neurological functions in NDDs depends on neurogenic differentiation, cell replacement, and the secretion of neurotrophic factors [191]. Unfortunately, the direct transplantation of MSCs at the injury site or injection into the vascular system frequently translates, within several days, into their death, due to natural senescence [192], the hostile microenvironment, and (or) nutrient deprivation [193]. Thus, even if the transdifferentiation of MSCs into neurons provides a practical technique for NDD treatment, it is limited by the unmet challenge of getting well-differentiated and mature neurons. The use of a single or combination of growth factors has been adopted to guide the differentiation of MSCs into a neuronal lineage. Growth factors, including Epidermal Growth Factor (EGF), Fibroblast Growth Factor, basic (bFGF), and Platelet-derived Growth Factor (PDGF), engage various cell surface receptors and a variety of signaling pathways, which often crosstalk, leading to an unexpected biological outcome [194]. Moreover, they give a faster differentiation rate often associated with transient changes in the gene expression profile with morphological changes, but without a clear distinction of neuronal functionality [195]. Thus, most differentiated MSCs in vitro adopt a neuron-like morphology. The use of electrophysiological assays can demonstrate that induced neuron-like MSCs have neuronal electrophysiological properties, including the capacity to evoke action potentials, and to respond to several neurotransmitters, such as GABA, glycine, and glutamate [196]. However, only a limited number of studies have carried out an electrophysiological analysis, which is important to demonstrate that MSC-derived neuron-like cells can also exhibit neuronal functions. This is necessary to consider these cells clinically relevant for use in neural repairs.

New strategies for the differentiation of MSCs into neurons, which could eventually be used to treat patients who are in need, are recently developing. These include (i) highly specific systems for MSC differentiation into neurons directed by local electrical stimuli [30], and (ii) MSC-based gene delivery strategies [197]. Since the inherent characteristics of neurons is to transmit electrochemical signals throughout the nervous system, electrical stimulation could significantly promote the neural differentiation of stem cells or neuron maturation [198,199], with different potential advantages, such as rare immune response, controllable parameters, low damage, easy implementation, localized induction, and synergy with other inducers. Thus, most studies have used external electric fields generated by electrodes or large electrical signal-generating devices to directly induce stem cell differentiation [200,201]. However, this is an invasive approach with an increased risk of wound pain and infection, unsuitable for nerve repair in humans [202]. Thus, recently, increasing research has focused on the development of an implantable, low-cost, non-invasive wireless stimulation system [203,204,205,206] and on the use of stimulus-responsive materials, such as graphene, which has been shown to promote MSC neural differentiation [207]. Today, this strategy has led to the development of a system that allows for MSC differentiation on carbon nanotube membranes stimulated with wireless electrical signals. Neuron-specific protein and gene analysis has shown that MSCs cultured on carbon nanotube membranes could differentiate into neurons under wireless electrical stimulation without any neural directing factor. Interestingly, MSC-derived cells have shown electrophysiological properties of neurons. This wireless electrical stimulation system can induce in vivo MSC differentiation (nearly 100% of neurons without astrocyte cells), thus providing new avenues for autologous stem cell therapy in NNDs, non-invasive nerve repair, and cell regeneration [30]. Although micro-fabrication is effective and precise, the high cost and complicated process limit the clinical application of this promising non-invasive electrical stimulation system to induce in vivo MSC differentiation.

The transfection of MSCs with genes that promote cell resistance to hypoxia/ischemia, oxidative stress, and acute or chronic inflammation or with genes enhancing neurotrophy and neuroprotection may increase cell survival in vivo and, importantly, facilitate neuronal replacement and repairing, and the reconstruction of neural circuitry, thus potentially restoring neurological function [197]. The main tools for gene delivery include viral-based methods, which allow for the construction of stably transfected MSCs, with a more sustained gene expression time, and nonviral-based methods, always sustaining only a transient gene expression in MSCs [197]. In addition to the stable long-term expression of integrated genes, virus-based methods for gene delivery also have the advantage of a high infection efficiency, but potential disadvantages that can also reduce safety include immunogenicity, the risk of gene integration and the insertion of mutations, and lethal and carcinogenic risks. On the other hand, non-viral-based methods, which include physical methods (such as sonotransfection and electroporation) are characterized by a high transfection efficiency but they can be associated with high cytotoxicity and a lack of targeting. Moreover, non-viral-based methods can be difficult to apply in vivo [197]. Before their application in clinical practice, the biggest concern about genetically modified MSCs is their safety. Several clinical trials have significantly reduced concerns about the serious or life-threatening adverse events associated with natural MSC administration, but this does not guarantee the safety of genetically modified MSCs. The gene expression profiles of modified MSCs can dramatically change, thus bringing concerns about their safety. The safety of gene modification strategies that promote cell proliferation or survival requires more attention because in many tumor tissues, the expression of these genes is also abnormal [197]. Up until now, there still have been inherent challenges in both the clinical translation and application of these methods. In particular, it will be difficult for genetically modified MSCs to enter clinical trials until some specific issues are resolved, such as the selection of more effective and safer genes, the potential risk of tumors or other diseases associated with genetically modified MSCs, and the risks to humans associated with the use of viruses for viral-based methods.

## 7. Immunological Response in Cell Therapy for Neurodegenerative Diseases

As discussed above, the successful application of cell therapy as an innovative strategy to replace damaged neurons in NDDs, such as PD and HD, depends on multiple factors; among them, the host immune response is of paramount importance.

The well-documented immunomodulatory and regenerative properties of MSCs are the reason why they are being used for the treatment of many diseases, including NDDs. Moreover, as reported before, they have been considered “immune-privileged cells”, as they do not activate aggressive immune responses. For this reason, MSC treatments are performed without considering the histocompatibility and without preventing possible immune rejections [208]. However, several studies have provided evidence that mismatched MSCs are immunogenic: mismatches in HLA antigens between donor and recipient lead to serious complications such as graft failure, transplant rejection, or graft versus host disease (GVHD). Moreover, when MSCs are exposed to a pro-inflammatory environment or during the MSC differentiation process, the surface immunogenicity of these cells is increased [209]. Recently, it has been shown that MSC treatment may provoke donors’ humoral and cellular immune responses. This occurs mainly in allogeneic transplants. The production of donor-specific antibodies (DSA) in the serum of transplant recipients supports the alloantigen recognition by B cells [210]. It is plausible that the generation of DSA is the result of the indirect recognition of MSC HLA by patient antigen-presenting cells (APC) to CD4+ T cells. This causes the induction of allogenic-specific T CD4+ cells, which will activate HLA-specific IgG-producing B cells. Up until now, only a few studies have addressed the aspects of the safety or tolerability of allogenic MSC therapy to their immunogenicity. Recently, Sanabria de la Torre and colleagues showed that no correlation between alloantibodies and the safety of allogenic MSC therapy occurs [209]. However, only a limited number of the studies that they analyzed, in which allogenic MSCs were used as treatment, measured the production of alloantibodies. This does not allow to extract firm conclusions about the immunogenicity of allogenic MSC treatment and more studies are needed.

Several critical factors could impact the immune response and should be taken into consideration when implanting cells to treat neurodegeneration [208]. The first is the transplantation procedures: although these are becoming minimally invasive and extremely accurate, immunosuppression is needed to overcome the inflammation and morbidity associated with the procedure. However, immunosuppression therapy could cause toxicity and worsen the clinical scenario; thus, it should be accurately selected and monitored [211,212,213]. Other factors include the cell type used [fetal tissue, ESCs, iPSCs, neural progenitor cells (NPCs), MSCs], the presence of genetic modifications, and the degree of mismatch between the donor and recipient. The compatibility of the major histocompatibility complex (MHC), known in humans as human HLA, represents an important factor: the degree of mismatch between donor and host increases the risk of immune rejection, ranging from the absence of rejection to the need for immunosuppressive therapy throughout the lifespan. MSCs seem to be more compatible with the host’s immune system due to their low levels of MHC I and the lack of MHC II molecule expression [214,215]. Additionally, evidence suggests that the site of transplantation plays a crucial role in the success of the therapy [216]. Finally, the neuroinflammation process observed in NDD should be considered. The activation of astrocytes, microglia, and other immune mediators occurring in neuroinflammation could affect the survival of transplanted cells, their integration, and rejection [208]. The accurate evaluation of the effects of all these factors requires the standardization of immune-related aspects following cell transplantation. Moreover, the administered immunosuppressive therapy and its adverse effects, as well as the analysis of neuroinflammatory biomarkers, should be considered for an accurate evaluation of clinical outcomes in cell therapy for NDD.

Figure 2 summarizes the main strategies to restrain the immunological response following cell therapy for NDD. Among them, the graft of autologous material from an identical donor twin is associated with the lowest immunogenic risk. However, currently, obtaining this type of transplantation for patients with NDD, such as PD or HD, is not easy. Possible realistic alternatives have been proposed [208]. Among them, the selection of the donor based on HLA compatibility with the host, which has to be accompanied by treatment with immunosuppressive drugs, has been proposed; in this context, the generation of cell banks could increase the availability of HLA-matched cells [217,218,219]. Moreover, it is possible to enhance the immunomodulatory and overall therapeutic efficacy of stem cells by preconditioning (priming) them with growth factors and pro-inflammatory cytokines (including IFN-γ, TNF-α, FGF-2, IL-1α, and IL-1β), hypoxia, pharmacological drugs, such as valproic acid, all-trans retinoic acid, and other molecules including lipopolysaccharide or cathelicidin [220]. The priming approaches of MSC still have many limitations in their clinical translation. These include the induction of immunogenicity, high costs, the variability depending on the MSC tissue source and donor, and the lack of good manufacturing practice (GMP) grade certification for their clinical use. Finally, whether priming approaches may affect the long-term tumorigenic potential of MSCs has not been yet addressed. The use of hypoimmunogenic cells with deleted MHC-I and MHC-II genes and the increased expression of the surface protein CD47, considered a “don’t eat me” signal [221], causes the loss of cell immunogenicity when MHC class I and II genes are inactivated and CD47 is over-expressed, by retaining pluripotent stem cell potential and differentiation capacity [222]. However, hypoimmunogenic cells eluding immune monitoring may pose long-term risks of uncontrollable malignant transformation or impaired virus clearance.

Tolerance induction approaches include the blockade of co-stimulatory molecules that are crucial for T-cell activation, such as CD28-CD80/86 and CD40-40L [223]. However, the immune tolerance strategy mainly developed in mouse models must be re-evaluated in the context of the human immune system. The use of stem cells in combination with the target neuronal cells to immunomodulate the response upon grafting has also been proposed. In animal models, this approach can delay allograft rejection and preserve the functionality of the graft [224,225,226]. However, further study should be performed to improve the viability of the transplanted cells while also limiting the risk of unwanted cell proliferation. Finally, a promising strategy could be the encapsulation of cells using biocompatible carriers to obtain neuroprotective and neurodegenerative effects [227]. This approach has the advantage of combining biomaterials and cell therapy, which has already reached clinical trials.

In conclusion, up until now, the occurrence of the immune response when considering cell therapy for NDDs has remained an open challenge. The immune response may impair the survival of grafted cells and, therefore, their functionality. Thus, immunosuppression is needed to overcome the inflammation and morbidity associated with the procedure, and this could cause toxicity and worsen the clinical scenario. Most clinical assays in the field are not performed based on clear and feasible guidelines for monitoring the immune response. Moreover, the immunosuppressive treatments used now are highly variable, and the associated adverse effects are not always described. The accurate monitoring of the immunosuppressive treatment administered, their adverse effects, and the analysis of neuroinflammatory biomarkers should be considered for the accurate evaluation of clinical outcomes and an adequate balance between immunosuppression to avoid rejection and maintaining immune function.

## 8. Large-Scale Production of Human Mesenchymal Stem Cell Manufacturing for Clinical Uses

The clinical uses of MSCs are limited by technical problems associated with mass production, high manufacturing cost, and contamination. The production of MSCs on a large scale is further complicated by the need for manufacturing processes able to provide a high therapeutic quality and purity of cells according to the current GMP standards. Several expansion methods to obtain appropriate numbers of cells with preserved therapeutic quality have been proposed [228]. However, currently, an ideal method for the expansion of MSCs on a large scale remains an important challenge. Table 8 summarizes the current knowledge about manufacturing techniques for the large-scale production of GMP-grade MSCs [228].

The most used approach for the large-scale manufacturing of MSCs for clinical use is represented by standard bioreactor systems [229]. This automatic system of cell cultures allows for the growth of large numbers of adherent cells, providing reduced labor costs and improvements in cell quality, a central issue when scaling up the processes. Bioreactors can enable the frequent feeding of the culture; thus, they maintain the levels of metabolites necessary for cell expansion under control and allow for a faster and safer expansion of MSCs compared to conventional cultures [230].

In this system, the main process parameters to be controlled include temperature, pH, pO2, pCO2, microcarrier suspension, and shear stresses [228]. Therefore, it is necessary to develop online control systems that ensure that product characteristics remain unchanged. Another major limitation to the therapeutic use of MSCs is the composition of culture media, which hinders the validation of GMP-compliant processes. To date, a large number of laboratories use culture media supplemented with fetal bovine serum (FBS) to expand MSCs, but this option will be not applicable in the future. FBS has a not well-defined composition, and it may promote interspecies cross-contamination. Proposed alternatives include human platelet lysate (hPL), but the potential risk of disease transmission and its limited availability reduce its application to large-scale production. Alternatively, new GMP-compliant, commercially available, chemically well-defined xenogeneic-free media that support MSC growth would constitute a more cost-effective and risk-reduced approach. However, some changes in morphology, phenotype, potency, and cellular senescence have been reported, thus suggesting that methods for MSC culture need to be further optimized to enhance batch-to-batch consistency in the cell manufacturing process [231].

The challenge of the future in the field of the MSC manufacturing process is to harmonize it for different clinical conditions and to work to obtain a unique ‘off-the-shelf’ MSC product. Ideally, this product should be derived from freely available tissue sources, such as umbilical cord tissue, which can be collected with non-invasive procedures. Moreover, it should be cultured in GMP- and regulation-compliant xenogeneic-free media and expanded in a closed automated bioreactor system. Then, it should be delivered ‘off-the-shelf’ as a cryobanked product suspended in a chemically defined, dimethyl sulfoxide (DMSO)-free media. DMSO is an efficacious and economical cytoprotective agent, but its use can be associated with negative effects on humans depending on concentration, administration, and dose. Finally, the final cryobanked product should not require further manipulation at the bedside. Nevertheless, there is the need to repeat pre-clinical safety and efficacy studies when changes are introduced into the bioprocess.

As reported above, in recent years, several strategies have been designed to improve the therapeutic potential of MSCs, which, in Europe, are considered advanced therapy medicinal products (ATMPs). However, the manufacture and handling of these cells for their use as ATMPs is still poorly studied, and a large part of the available data is not related to industrial processes. Up until now, the MSCs used to obtain ATMPs could only be isolated in authorized centers with processes which are standardized around the world. In contrast, the optimal protocols for culturing isolated MSCs are not standardized. This constitutes a major open challenge to improve their therapeutic properties. Cell culture conditions, such as the cell density, time of culture, and culture medium composition represent bottlenecks that need critical controls [230].

The quality of MSC products has to be ensured starting from the selection of primary materials and their packaging. Moreover, the control of the manufacturing process and testing of the quality of the final products has to be performed. The quality of MSCs is assessed by measuring purity, potency, and safety and it is crucial for the success of their therapeutic administration. These tests are performed with extensively standardized procedures but, especially when cells are long-term cultures, a karyotyping test and/or array-CGH is recommended to demonstrate the genetic stability of the final cell product. Microbiological tests must be performed before packaging at the end of the manufacturing procedure [230].

In the future, cell therapy might become predominant in medicine, but only some centers will be able to produce and supply cells and develop procedures to control these cell shipments. Any mistake in this process could alter the final product, thus compromising its therapeutic use. The selection of excipients, storage and shipping temperature, and container conditions are particularly important. The correct handling of the cells is critical for the success of cellular therapies and its manipulation and administration require special care and training. To date, both public institutions (including academia and hospitals) and private enterprises are interested in the development of novel ATMPs. However, academic institutions, and also small enterprises, experience great difficulty in moving to the advanced phases of the studies mainly due to a lack of personnel, infrastructure, and capital. Contrastingly, large private companies may have more resources for scale-up strategies, but they may consider a high-risk approach to invest in novel cell-based therapeutics for low-incidence diseases or if a complete understanding of the product’s mechanism of action is still lacking. Through a partnership model between these two sectors, it would be possible to share the knowledge, skills, and resources, but also the risks associated with ATMP development.

## 9. Conclusions and Future Perspectives

Cell-based therapies have been proposed as a promising tool in the treatment of several human NDDs. Remarkably, preclinical studies have demonstrated encouraging results on the functional benefit of cell transplantation in the neurological field and several efforts have been undertaken to successfully apply cell therapy in NNDs. However, to date, this approach has remained experimental and most of the undertaken clinical trials have not been properly designed to assess efficacy and to confirm the promising results about the safety. One of the challenges when considering cell therapy for NNDs is the immune response, which can compromise the survival of grafted cells, impairing their integration and, therefore, their functionality. For this reason, in recent years, in vitro strategies have been developed to evaluate the potential immunogenicity of cell therapy. Funding agencies and the neuroscience community should invest in this kind of strategy to improve and standardize preclinical studies for the development of cell therapies. Good well-standardized in vitro models may provide access to understanding some mechanisms that are difficult to assess in animal models. In fact, the molecular mechanisms underlying the beneficial effect of cell therapy are still largely unidentified. There are some concrete pieces of evidence to support the hypothesis that the transplanted cells produce neurotrophic factors, enhance neuronal plasticity, and activate local progenitors and cell replacement (Figure 3).

So far, MSCs have been largely investigated for their differentiation capacity, paracrine effects, and direct-contact modulatory functions. Each mechanism contributes to the comprehensive process of MSC therapy. Importantly, MSCs can adapt therapeutic effects during the rescue and repair of damaged tissues according to diverse local microenvironments. Therefore, the in-depth mechanisms underlying the protective effects of MSCs deserve further investigation. The clarification of the predominant mechanisms in different situations will improve the safety, efficacy, and outcomes of MSC-based therapy.

Despite the promising results, there are several challenges to overcome to translate stem cell therapies from the bench to patients. Among them, the optimal timing, dosage, and route of administration should be accurately determined. For example, most clinical trials use doses considerably below this efficacious threshold, which may explain the observed lack of efficacy. In addition, considerations regarding the administration of single or multiple doses, as well as the timing of these injections, can also result in significant differences in outcomes. Dose standardization is crucial to reduce variability between trials and gain insights into areas that require improvement. The optimal route for stem cell injection can vary depending on the nature of the disease. Variability in the outcomes of clinical trials importantly contribute to the translatability of preclinical findings to human trials. A significant source of this variability is represented by the heterogeneity of patient populations. The main factors promoting this variability include disease onset, development, pathological phases, disease severity, and the presence of co-morbidities. Therefore, careful consideration must be given to the rational selection of optimal candidate patients.

Another important contribution to the differences in the response rates between pre-clinical studies and clinical trials with MSCs may be due to the main use of fresh culture-derived MSCs in the pre-clinical studies versus the predominant use of cryopreserved MSCs in clinical trials. Therefore, regulatory bodies, such as the Food and Drug Administration (FDA) and the European Medicines Agency (EMA) require the extensive testing of MSCs for safety and efficacy, thus making essential the conforming of cryopreservation methods of MSCs to the regulatory standards. A cryopreservation step in the manufacturing process is associated with important benefits since it allows for immediate off-the-shelf access to the products and also the completion of all quality testing before administration to the patient. Moreover, cryopreservation cannot be avoided in MSC banking strategies. However, the cryopreservation of MSCs may impair their functional properties as compared to freshly harvested MSCs. Thus, targeted research into optimizing the cryopreservation and freeze–thawing protocols may importantly contribute to increasing both the safety and efficacy of cellular therapeutics in clinical use and lead to effective, economically viable, and sustainable market strategies. In Table 9, we summarized the key aspects of cryopreservation and the freeze–thawing methods of the MSC manufacturing process and the suggested strategy to improve them for a better therapeutic outcome [231].

In conclusion, we have outlined and discussed the promising results of gestational stem cell transplantation in treating NDD (Figure 4).

Their potential clinical advantages are linked to (1) the large accessibility of these cells; (2) the availability of large quantities for transplantation; (3) the lack of ethical concerns; (4) the absence of tumorigenicity after transplantation; and (5) the ability to differentiate into multiple lineages. However, convincing evidence about the functionality status generated in differentiating neurons is still lacking. In recent years, these limitations have strongly promoted the development of new methodologies for the differentiation of neuroglia that could eventually treat patients who are in need, including highly specific systems for MSC differentiation into neurons directed by local electrical stimuli and MSC-based gene delivery strategies.

Thus, although several issues and limitations need to be addressed, stem cells represent a promising therapeutic approach in the clinical management of incurable diseases with the hope of prolonging and improving the human life of many patients.

## Figures and Tables

**Figure 1 ijms-25-00976-f001:**
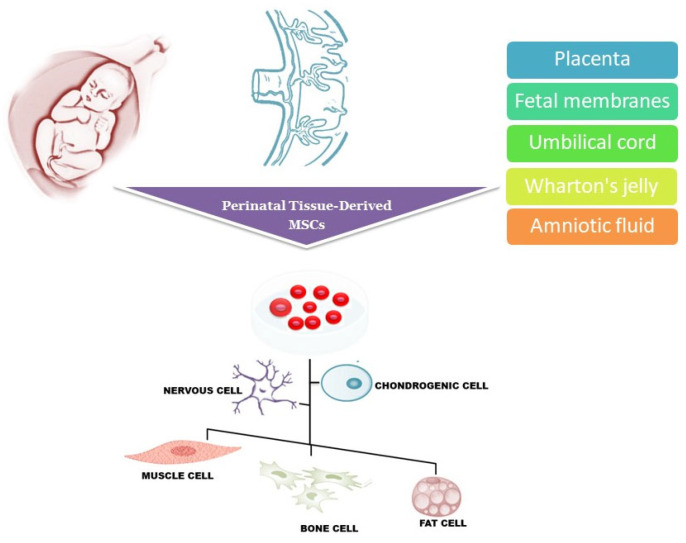
Perinatal tissue-derived stem cells, which can be isolated from placenta, fetal membranes, umbilical cord, Wharton’s jelly, and amniotic fluid, can differentiate into several cell lineages, including muscle, nervous, chondrogenic, bone, and fat cells.

**Figure 2 ijms-25-00976-f002:**
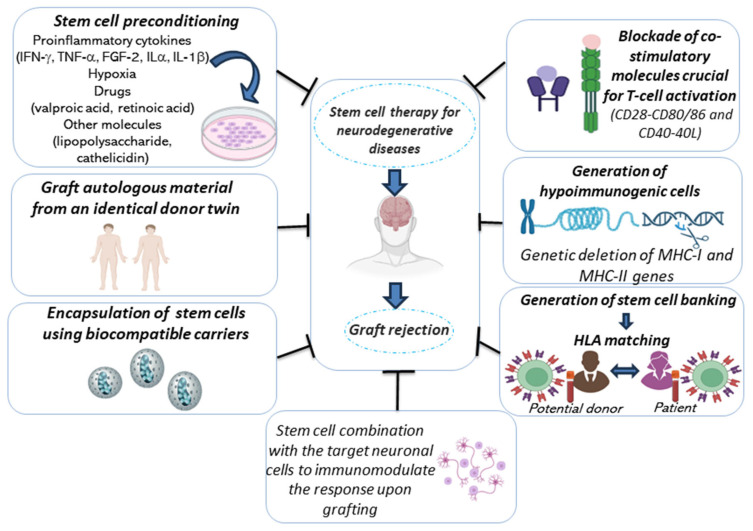
Strategies to overcome graft rejection in cell therapy for neurodegenerative diseases. Created in Biorender.com.

**Figure 3 ijms-25-00976-f003:**
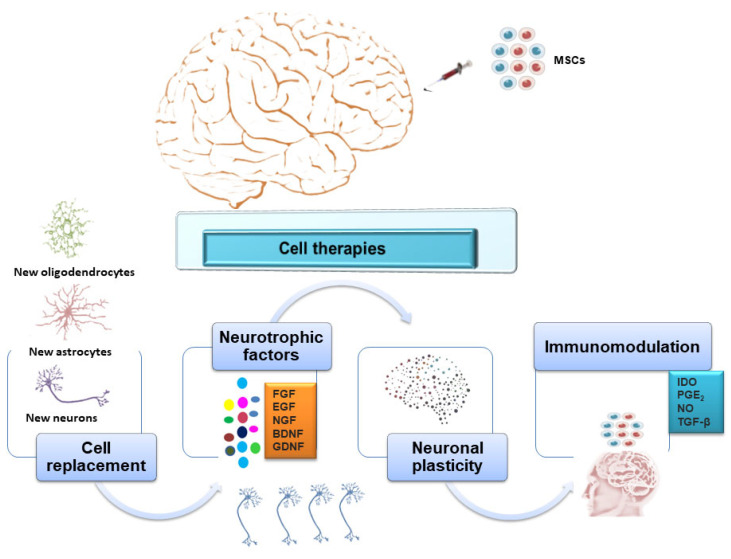
Mechanisms underlying the functional benefits of MSC transplantation in the neurological field can be associated with their capacity to activate local progenitors and cell replacement, to produce neurotrophic factors, including Fibroblast growth factor (FGF), Epidermal Growth Factor (EGF), nerve growth factor (NGF), brain-derived neurotrophic factor (BDNF), glial cell-derived neurotrophic factor (GDNF), enhance neuronal plasticity and the immunomodulation mediated by several factors, including Indoleamine 2, 3-dioxygenase (IDO), prostaglandin E_2_ (PGE_2_), nitric oxide (NO), transforming growth factor (TGF)-β these limitations.

**Figure 4 ijms-25-00976-f004:**
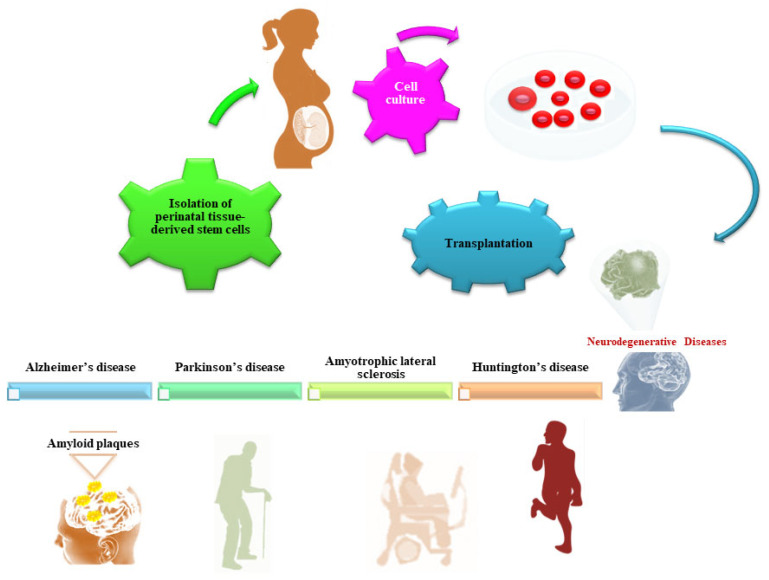
Potential use of perinatal tissue-derived stem cells in the treatment of neurodegenerative diseases (NDD) discussed in this review: Alzheimer’s disease (AD), Parkinson’s disease (PD), Huntington’s disease (HD), amyotrophic lateral sclerosis (ALS).

**Table 2 ijms-25-00976-t002:** Clinical trials on perinatal tissue-derived MSCs in Alzheimer’s disease.

Trial ID	Stage	MSC Type	Study Design	Route	Arms	Findings
NCT01297218	Completed	hUCB-MSCs	Phase 1 open-label, single-center	A single intracerebroventricular infusion	DOSE A—250,000 cells per 5 μL per 1 entry site, 3 million cells per brain DOSE B—500,000 cells per 5 μL per 1 entry site, 6 million cells per brain	hUCB-MSC injections into the hippocampus and precuneus by stereotactic surgery are safe and well tolerated but are not able to evoke the beneficial effects on AD pathophysiological process
NCT01696591	Unknown	hUCB-MSCs	Long-term safety and efficacy follow-up of NCT01297218		Same as NCT01297218	
NCT02054208	Completed	hUCB-MSCs	Phase 1/2a, double-blind, single-center	Ommaya Reservoir intraventricular injection	*n* = 42 Three injections at 4-week intervals Low dose: 1 × 10^7^ cells/2 mL High dose: 3 × 10^7^ cells/2 mL Placebo group saline	Repeated intracerebroventricular administrations are sufficiently safe and relatively well tolerated; repeated administrations are necessary to maintain the therapeutic effects
NCT03172117	Unknown	hUCB-MSCs	Long-term follow-up of NCT02054208 phase 1 and 2 (randomized quadruple blind controlled)		Same as NCT01297218	
NCT04040348	Active	Allogeneic hUCB-MSCs	Phase 1, prospective, open-label trial	I.V.	*n* = 6 4 doses of 100 million UCB-MSCs once every 13 weeks within a year period	N/A
NCT04684602	Recruiting	AM- and UC-MSCs	Multi-center, prospective, open-label clinical trial	Injection via condition-specific route of administration	*n* = 5000	N/A
NCT01547689	Unknown	hUC-MSCs	Phase ½, open-Label, single-Center, self control clinical trial	8 I.V. infusions	*n* = 30 0.5 × 10^6^ UC-MSCs per kg once every two weeks in the first month of each quarter Time interval: two and a half months	N/A
NCT02672306	Unknown	hUC-MSCs	Multicenter, randomized, double-blind, placebo-controlled trial	8 I.V. infusions	*n* = 16 Experimental group: 0.5 × 10^6^ UCMSCs per kg once every two weeks Placebo group: normal saline once every two weeks	N/A
NCT03899298	Not yet recruiting	AM- and UC-MSCs	Phase 1	I.V. infusion and intranasal procedure	*n* = 5000	N/A
NCT02899091	Unknown	P-MSCs	Phase I/IIa, randomized, double-blind, placebo-controlled clinical trial	1/2 I.V. infusions	*n* = 24 2.0 × 10^8^ cells on day 0 2.0 × 10^8^ cells on day 0 and on week 4 (repeated injection) Placebo on day 0 and on week 4 (repeated injection)	N/A

Human umbilical cord mesenchymal stem cells (hUC-MSCs); human amniotic membrane mesenchymal stem cells (hAM-MSCs); human umbilical cord blood-derived mesenchymal stem cells (hUCB-MSCs); umbilical cord-derived mesenchymal stem cells (UC-MSCs); amniotic membrane-derived mesenchymal stem cells (AM-MSCs); placenta-derived mesenchymal stem cells (P-MSCs); not available (N/A).

**Table 3 ijms-25-00976-t003:** Key findings from preclinical studies with transplantation of perinatal tissue-derived MSCs in Huntington’s disease murine models.

MSC Type	Cell Number	Passage Number	Mouse Model	Route of Transplantation	Key Findings	Ref.
UC-MSCs	200,000	P3 to P8 or P40 to P50	R6/2 and WT mice	intrastriatal	Transient behavioral sparing in transplanted mice; Neuropathology in high-passage UC-MSCs transplanted mice (−) (↓); Release of trophic factors and immunomodulating cytokines (+), behavioral deficits (−)	[142]
UC-MSCs	250,000	P4	3-nitropropionic acid-lesioned mouse	bilateral striatal	Survival and migration of grafted MSC (+); Striatal volume and mean dendritic length of the striatum (+)	[129]
CM-hAMSC	500,000	-	R6/2 mouse	intraperitoneal	Clasping reflex (↑); Motor coordination and activity (↑); Striatal neuropathology (↑); Neuroprotection against inflammation	[114]

Umbilical cord mesenchymal stem cells (UC-MSCs); conditioned medium from mesenchymal stem cells isolated from the amniotic membrane (CM-hAMSC); (↑) improvement of disease signs; (+) increase; (−) reduction.

**Table 4 ijms-25-00976-t004:** Key findings from preclinical studies with transplantation of perinatal tissue-derived MSCs in Parkinson’s disease murine models.

MSC Type	Cell Number	Passage Number	Mouse Model	Route of Transplantation	Key Findings	Ref.
hpMSCs	2 × 3.2 × 10^5^	P3	PD rat model	Injection into two sites in the striatum	Rotational asymmetry (↑); Dopaminergic differentiation (+)	[152]
hUC-MSCs	1 × 10^3^	P9	PD rat model	Single stereotaxic injection	Number of rotations (−); Apomorphine-induced rotations (↑)	[155]
hUC-MSCs	1 × 10^6^	-	Hemiparkinsonian rodent model	Injection in the corpus striatum	Number of rotations (−); The migration of the hUC-MSCs in the lesioned brain tissue; Differentiation into neuron-like cells; Homing capability of hUC-MSCs to the lesioned brain tissue	[156]
hUVMSC-derived dopaminergic-like cells	1 × 10^6^	P2-P5	PD rat model	Stereotaxic injection into the rat striatum	Integration of the transplanted cells into the striatum (+); Dopamine levels in the brain tissue (+); Lateral rotations (−); Behavioral deficit (↑)	[159]
hUC-MSCs	1 × 10^6^	-	PD rat model	Stereotaxic injection into the substantia nigra	Apomorphine-induced rotations (↑); Improvement of the rotation; Expression of human nuclear-specific antibody (HNA) and Tyrosine-Hidroxylase (TH)	[160]
hUVMSC-derived dopaminergic-like cells	1 × 10^5^	P3	PD rat model	Stereotaxic injection into the rat striatum	Post-transplantation survival of differentiated hUVMSC (+); Expression of TH in rat brain; Apomorphine-induced rotations (↑)	[161]
hAMSCs	-	-	PD rat model	Unilateral supranigral injections	Survival of hAMSCs transplanted cells Protective effect of hAMSCs on dopamine neurons in substantia nigra of rats	[164]
hAMSCs	2 × 4 × 10^4^	-	PD rat model	Stereotaxic injection in the striatum	Recovery of apomorphine-induced rotational asymmetry (↑); Survival of hAMSCs in the striatum	[162]
hp-MSC-derived neural phenotype cells	2 × 1.5 × 10^5^	-	PD rat model	Unilateral injection into right medial forebrain bundle	Rehabilitation of motor deficit; Loss of dopaminergic neurons (−); Protection of DA neuron loss; Survival of cells transplanted and differentiation into neurons at the grafted sites; Expression of DLL-1, MASH1 and NRTN (+); Modulation of immune responses of damaged brains; Neuroprotection and the inhibition of neuroinflammation	[166]
hUC-MSC-derived dopaminergic-like neurons	1 × 10^6^	P3	PD rat model	Stereotaxic injection in the right striatum	Apomorphine-induced rotations (↑); Expression of GFAP, Iba-1, TH, Nurr1, Pitx3, BDNF, TrkB, PI3K, p-Akt, Hsp60, TLR4, and MyD88 in striatum (+); Inhibition of neural apoptosis in the substantia nigra and striatum; Possible mechanism of neuroprotection and antineuronal inflammation	[167]
WJ-MSCs	2 × 1 × 10^6^	-	PD rat model	Injection into medial forebrain bundle	Motor activity (↑); DA and TH concentration (+);	[168]

Human placenta mesenchymal stem cells (hpMSCs); human umbilical cord mesenchymal stem cells (hUC-MSCs); mesenchymal stem cells from the amniotic membrane (hAMSCs); human umbilical vein mesenchymal stem cells (HUVMSCs); human placental-derived mesenchymal stem cells (hP-MSCs); Wharton’s jelly mesenchymal stem cells (WJ-MSCs); (↑) improvement of disease signs; (+) increase; (−) reduction.

**Table 5 ijms-25-00976-t005:** Clinical trials on perinatal tissue-derived MSCs in Parkinson’s disease.

Trial ID	Stage	MSC Type	Study Design	Route	Arms	Findings
NCT03684122	Active, not recruiting	hUC-MSCs	Phase 1 Phase 2	intrathecal and intravenous injection	-	N/A
NCT05691114	Recruiting	hAESCs	Phase 1	Ommaya reservoir implanted into the lateral ventricle	Dose A (5 × 10^7^ cells/dose) Dose B (1.0 × 10^8^ cells/dose) Dose C (1.5 × 10^8^ cells/dose).	N/A
NCT04414813	Completed	hAESCs	Early phase 1	Stereotactic transplantation of hAESCs	Transplantation of 50 millions hAESCs	N/A
NCT05435755	Not yet recruiting	hAESCs	Early phase 1	Precise transplantation of hAESCs into the ventricle	A total of 6 hAESC transplants will be performed. A total of 50 million (in 2 mL) hAESCs will be transplanted into the ventricle of participants through the Ommaya sac (set as day 0 at the beginning of the trial). Subsequently, hAESC ventricle transplants will be performed at 1 month ± 5 days, 2 months ± 5 days, 3 months ± 5 days, 6 months ± 5 days, and 9 months ± 5 days after the first cell transplantation, with a volume of 50 million cells (in 2 mL) each time.	N/A
NCT04876326	Unknown	hUC-MSCs	Not applicable	Allogeneic umbilical cord mesenchymal stem cell implantation	Implantation of autologous mesenchymal stem cells with origin of adipose tissue with a dose of 2 × 50 million cells given with an interval of 1 month	N/A

Human umbilical cord mesenchymal stem cells (hUC-MSCs); human amniotic epithelial stem cells (hAESCs); not available (N/A).

**Table 6 ijms-25-00976-t006:** Key findings from preclinical studies with transplantation of perinatal tissue-derived MSCs in amyotrophic lateral sclerosis murine models.

MSC Type	Cell Number	Passage Number	Mouse Model	Route of Transplantation	Key Findings	Ref.
hUCBCs	100,000	P4	ALS mouse model	Intrathecal stem cell transplantation	Beneficial effects on pre-symptomatic motor performance (↑); Considerable amounts of transplanted cells within the CNS	[172]
hUCBCs	1 × 10^6^	-	Transgenic mouse model	Intraspinal injection	Survival time (+); Motor performance (↑) Motor neuron loss and astrogliosis in the spinal cord (−);	[174]
hUCBCs expressing VEGF	1 × 10^6^	-	ALS mouse model	Retro-orbital injection	Integration of HUCB transplanted into nervous tissue of ALS mice; Survival for over 3 months; Migration of modified HUCB cells in the spinal cord parenchyma, proliferation, differentiation into endothelial cells; Neuroprotective effects	[175]
SPIO-labeled hUC-MSCs	4 × 10^5^	P5	Adult rat model	Injection into the dorsal spinal cord	Extensive human cell survival and engraftment within the injured rat spinal cord (+); Promotion of locomotor recovery (↑); Migration in the host spinal cord after transplantation	[176]
SPIO-labeled hUCBCs	100,000	-	ALS mouse model	Intraspinal injection	Detection of SPIO-labeled hUCBCs in ALS model by MRI; No migration of the injected cells or dislocation in the spinal cord	[177]
hAFCs	100,000	-	PD rat model	Transplantation in the lateral ventricles of wobbler	Mortality rate (−); Absence of inflammatory response and rough modification of the ventricular system	[178]
hAMSCs	1 × 10^6^	P6–P8	ALS mouse model	Intravenous administration	Delay of the disease progression and extension of survival time (↑); Motor performance on the rotarod (↑) Prevention of weight loss (↑); Neuroinflammation (−); Motor neuron loss in spinal cord ventral horns (−); Microglial activation (−)	[180]

Human umbilical cord blood cells (hUBCs); human umbilical cord (hUC) blood-derived cells; human umbilical cord mesenchymal stem cells (hUC-MSCs); human amniotic fluid cells (hAFCs); mesenchymal stem cells from the amniotic membrane (hAMSCs); (↑) improvement of disease signs; (+) increase; (−) reduction.

**Table 7 ijms-25-00976-t007:** Clinical trials on perinatal tissue-derived MSCs in amyotrophic lateral sclerosis.

Trial ID	Stage	MSC Type	Study Design	Route	Arms	Findings
NCT04651855	Active, not recruiting	WJ-MSCs	Phase 1 Phase 2	Intrathecal administration	Three times for each enrolled patient	N/A
NCT05003921	Suspended (modifying protocol)	AlloRx UC-MSCs	Phase 1	Intrathecal injections	Three intrathecal injections of 50 million cells at two-month intervals	N/A
NCT01494480	Unknown status	UC-MSCs	Phase 2	stem cell transplantation	Four stem cell transplantations through lumbar puncture, the time is 3–5 days between two treatments	N/A

Wharton’s jelly mesenchymal stem cells (WJ-MSCs); umbilical cord mesenchymal stem cells (UC-MSCs); not available (N/A).

**Table 8 ijms-25-00976-t008:** Manufacturing techniques for large-scale production of GMP-grade hMSCs.

Manufacturing Technique	Description	Main Limitations
Two-dimensional culture	Use of multilayer vessels with multiple cell growth chambers stacked within the unitary flask body that could be held in the incubator	Not suitable for large clinical trials (>100 patients) and for allogeneic transplantation Open culture system with increased contamination risk. Occurrence of morphological, phenotypic, genetic, and epigenetic changes affecting cell fate, treatment efficacy, and patient safety
Spinner flasks	Easy-to-use, closed, and dynamic cell culture system	Inability to control additional critical process parameters, such as pH and dissolved oxygen High price of automation
Roller bottles	Easy-to-use, closed, and dynamic cell culture system with a larger yield of cell growth	Not effective in the production of a significant number of hUC-MSCs for therapy purposes. High price of automation
Three-dimensional culture	Cells directly interact and spontaneously adhere to each other, leading to the formation of multicellular aggregates. More similar to physiological conditions compared to 2D cell culture	Aggregate formation increases survival, viability, and proliferation of MSCs on the surface; however, oxygen and nutrient gradient in the core induce a high number of quiescent or necrotic cells
Standard bioreactor systems	Completely closed, manageable, and scalable culture technique that could allow for large-scale and GMP-compliant hMSCs manufacturing. Precise monitoring and tight regulation of culture parameters such as pH value, temperature, dissolved O_2_ and CO_2_ level. A homogeneous physicochemical environment in culture medium has a profound impact on maintenance of hMSCs potency and therapeutic efficacy	Culture conditions are characterized by simplified geometric properties and matrix composition, as well as reduced intercellular communication, compared to tissue microenvironment

**Table 9 ijms-25-00976-t009:** Optimization strategies of key aspects of cryopreservation and freeze–thawing methods of the MSC manufacturing process.

Key Aspect	Current Approaches	Comments	Optimization Strategy
Definition of cryoprotectant compositions	Dimethyl sulfoxide (DMSO)	Low-cost. Great efficacy as a cytoprotective agent. Negative effects on humans depending on concentration, administration, and dose.	Supplementation of 10% DMSO with a buffer containing reagents ranging from 5% Human Albumin, Human Serum, or Human Plasma A/B. More complex formulations involving Dextran-40, Lactobionate, Sucrose, Mannitol, Glucose, Adenosine, or Glutathione
Container for cell cryogenization	Bags for cell cryogenization. Standard screw cap Cryotube (for small volumes)	Not suitable for procedures under GMP, since the closing systems are not safe and can favor pollutant entry	Small-volume cryopreservation systems are currently being developed in a completely closed system
Storage for long periods	Storage at a temperature below −120 °C, usually in the gaseous phase of liquid nitrogen	Liquid nitrogen could transmit contamination from one cryo bag or cryotube to another	Use of nitrogen tanks suitable for their function with clearly differentiated compartments (i.e., racks) to store the different batches without loss or cross-contamination. Use record forms to ensure the traceability of the cells. Use of a qualified storage temperature recording system. Use of controlled-rate-freezing devices to optimize the freezing rate and prevent ice crystal damage of cells. Limited access to the cryopreservation unit to authorized personnel only. Cellular stocks (small cell banks)
Characterization and delivery of thawed MSCs to patients	Culture recovery of cryopreserved MSCs for 1–2 days before in vivo use to restore optimal cellular function. Stimulation of MSCs with IFN-γ before initiating the cryopreservation process to maintain MSC immunomodulatory activity post-transplantation	Before clinical application, cells should be cultured for a passage, although other available protocols also provide optimal therapeutic potential	Post-thawing release criteria should include parameters such as viability, recovery, phenotyping, and potency assay. Viability and engraftment in cell delivery could be improved through cell encapsulation, hydrogels, and biomaterial-assisted approaches

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
