# Peer review of "Perinatal Tissue-Derived Stem Cells: An Emerging Therapeutic Strategy for Challenging Neurodegenerative Diseases"

_ijms, 2024, doi:10.3390/ijms25020976_

Round 1

Reviewer 1 Report (Previous Reviewer 1)

Comments and Suggestions for Authors

The authors had made satisfactory edits in the manuscript, however, the statement in the cover letter about electrophysiology is worrisome and reflect exactly the issue in the field about the transdifferentiation of MSCs in neuron-LIKE cells:

Only few study shows the capabilities of newly formed nerve cells to carry electrical charges, by performing electrophysiological analysis; however, this analysis is not able to demonstrate the formation of functional synapses.

This statement is completely misleading and incorrect as this analysis will demonstrate what only neuronal cells can perform. It is true that few studies are performing it which emphasize this kind of statement and the idea that morphology and neuronal marker are enough. As long as the authors understand and state that without electrophysiology, the differentiation of MSCs can only be characterized as neuronal-like cells (which appears in the manuscript -line 77-80). This reviewer would be enclined to accept the manuscript.

Author Response

Reviewer 2 Report (New Reviewer)

Comments and Suggestions for Authors

The manuscript titled "Perinatal tissue-derived stem cells: an emerging therapeutic strategy for challenging neurodegenerative diseases" is a notable contribution to neurodegenerative disease research. It adeptly explores the use of perinatal tissue-derived stem cells, shedding light on their potential in treating conditions such as Alzheimer's, Parkinson's, Huntington's, and Amyotrophic Lateral Sclerosis.

This work stands out for its comprehensive analysis of the scientific and clinical aspects of stem cell therapy. It delves into the potential mechanisms of action of these cells, including neurotrophic factor production and enhancement of neuronal plasticity. The manuscript also critically addresses the challenges in translating laboratory research into clinical applications, discussing issues like immune response, cell culture standardization, and production scalability.

While celebrating the therapeutic potential of these cells, the manuscript remains balanced, acknowledging the ethical and practical hurdles in their application. This nuanced approach makes the manuscript a valuable resource for both researchers and clinicians in the field. With some further refinement in discussing future research directions and emerging technologies, this work could have an even greater impact in guiding stem cell therapy in neurodegenerative diseases.

Here are some recommendations to improve the overall quality of your manuscript:

Sections 1, 2, and 2.1

1. The manuscript provides a comprehensive overview of the current state of research in the field of stem cell therapy for NDDs. However, it might benefit from a more detailed discussion on the limitations and challenges faced in translating these preclinical findings into clinical applications. For instance, issues regarding the scalability of MSC production, potential immune rejection, and long-term safety need to be addressed.

2. The section on AD is thorough but could be enhanced by including more recent findings or ongoing clinical trials if available. This would provide a more current perspective on the topic.

3. The manuscript is well-structured, with a clear progression from a general overview of NDDs to specific details about cell therapy in AD. However, some sections could be reorganized for better flow. For example, the part discussing the sources of MSCs could come earlier in the manuscript, providing a clearer background before delving into specific diseases.

4. The manuscript is generally well-written, but some sentences are overly complex and could be simplified for clarity. Avoiding long sentences with multiple clauses would make the text more accessible, especially to readers not intimately familiar with the topic.

5. There are instances where technical jargon is used without explanation. While the target audience is presumably knowledgeable in the field, defining or briefly explaining key terms at their first mention would be helpful.

6. The manuscript briefly touches on ethical considerations regarding stem cell use. Expanding this section could be valuable, addressing both the ethical implications and public perception of using perinatal tissues for medical research and therapy.

Section 2.2

7. The summary of clinical trials is comprehensive and provides a good overview of the current state of research in the field. However, it would be beneficial to provide a more critical analysis of these trials, discussing not only the findings but also the limitations and potential biases. For example, you could explore the implications of the small sample sizes, the lack of long-term follow-up data, or the variability in trial design.

8. The manuscript does a good job of highlighting the disparity between animal and human studies. It would be informative to delve deeper into why these differences might exist. For instance, consider discussing factors such as the complexity of human AD pathology compared to animal models, differences in the brain microenvironment, or issues with the scalability and reproducibility of cell therapy.

9. While the manuscript touches upon the clinical aspects of MSC therapy, there is a lack of discussion on the ethical and regulatory challenges. Addressing how these therapies are regulated, especially considering the use of gestational tissues, and discussing the ethical implications of such treatments could provide a more rounded view of the subject.

10. The interpretation of the clinical trial data seems to lean towards cautious optimism. It might be helpful to discuss more explicitly the clinical relevance of these findings. How do these results translate to potential treatments? What are the hurdles that still need to be overcome before these therapies can be considered viable for AD patients?

11. The manuscript could benefit from a section on future directions, outlining what steps are needed next in both research and clinical practice. Suggestions for improving trial design, such as incorporating larger and more diverse patient populations, or recommendations for further studies to address unanswered questions would be valuable.

Sections 3 and 4

12. The introductory part about Huntington’s Disease (HD) is concise and informative. However, it could be enhanced by providing a bit more detail on the genetic mechanism behind HD, especially for readers who may not be familiar with the disease. A brief explanation of how the CAG repeat expansion in the HTT gene leads to the disease pathology would be beneficial.

13. The section on the role of BDNF and GDNF in HD is well-presented. It would be helpful to include a discussion on how these factors specifically impact HD pathology and why they are considered potential therapeutic targets. Also, including a brief explanation of why previous studies have shown inconsistent results regarding BDNF levels in HD patients could provide more depth.

14. The manuscript does a good job summarizing various preclinical studies. However, it would be beneficial to provide a more critical analysis of these studies. Discuss the limitations of these studies, such as the challenges of translating findings from animal models to human patients, and the variability in study designs and methodologies.

15. The manuscript discusses various sources of MSCs, including bone marrow, adipose tissue, and gestational tissues. A comparative analysis discussing the pros and cons of each source in the context of HD treatment would provide a more comprehensive understanding for the readers.

16. While the manuscript mentions the potential mechanisms of MSC-mediated recovery in HD, it could delve deeper into this topic. Discussing how MSCs might interact with the neural environment in HD and the possible pathways through which they exert their effects would add valuable insight.

17. The manuscript would benefit from a concluding section that summarizes the current state of cell therapy in HD, highlights the main challenges, and suggests future directions for research. Discuss what needs to be addressed in future studies to move closer to clinical application.

Sections 5 and 6

18. The introduction to ALS and the potential of cell therapy is clear and concise. However, it would be beneficial to include a brief explanation of why ALS predominantly affects males, as stated, and whether this has implications for cell therapy approaches.

19. You have summarized various preclinical studies effectively. It would be helpful to delve deeper into the mechanisms through which transplanted cells exert their effects in ALS models. For instance, the role of neurotrophic factor delivery, immunomodulation, and cell replacement could be elaborated upon.

20. The manuscript mentions different cell types and transplantation techniques. A comparative analysis discussing the advantages and challenges of each type, such as hUC-MSCs, hAFCs, and hAMSCs, in the context of ALS therapy would be insightful.

21. The manuscript touches upon the combined effect of gene and stem-cell approaches. Expanding on this topic, such as detailing specific gene modifications that have been explored and their intended effects, would add depth to the discussion.

22. The manuscript mentions that only a few studies demonstrate the capabilities of newly formed nerve cells to carry electrical charges. Expanding on the importance of electrophysiological analysis in confirming functional integration of transplanted cells into neural circuits would strengthen this section.

23. The discussion on the use of non-invasive electrical stimulation for MSC differentiation is intriguing. Further elaboration on how these techniques compare to more traditional methods, and their potential advantages and limitations in clinical applications, would be beneficial.

24. The section on gene delivery strategies is informative. However, a more detailed discussion on the specific challenges in clinical translation, such as the safety concerns and the regulatory hurdles, would provide a more complete picture.

25. A concluding section that summarizes the current state of cell therapy research in ALS, highlights the main challenges, and suggests future research directions would be a valuable addition. This could include the need for larger, more diverse clinical trials or the exploration of combination therapies.

Sections 7 and 8

26. The section provides a comprehensive overview of the immunological challenges in cell therapy for NDDs. It would be beneficial to expand on the specific immune-mediated mechanisms that could lead to graft rejection. This includes detailing the roles of innate and adaptive immunity, and how different cell types may provoke varying immune responses.

27. Discussing the balance between immunosuppression to avoid rejection and maintaining enough immune function to not exacerbate disease symptoms or cause other complications would add depth to the discussion.

28. The strategies mentioned to overcome graft rejection are intriguing. A deeper dive into each strategy, elaborating on their practical implementation, potential risks, and current state of research would make the section more informative. Additionally, a discussion on the ethical and regulatory considerations of these strategies, especially those involving genetic modifications, would provide a more rounded perspective.

29. The section on manufacturing techniques for large-scale production of MSCs is well-structured. However, including a discussion on the scalability of these techniques, and how they can meet the demands of large clinical trials or widespread clinical use, would be beneficial. Addressing the challenges in maintaining cell quality, potency, and safety during scale-up processes, and the strategies to mitigate these challenges, would provide practical insights.

30. The manuscript mentions the use of standard bioreactor systems for MSC manufacturing. Elaborating on the technological advancements in bioreactor design, such as automation and real-time monitoring, and their implications for MSC production would be valuable. Discussing the economic aspects, such as cost-effectiveness and the feasibility of these systems in different settings (academic vs. industrial), would add a practical dimension to the discussion.

31. The discussion on culture media composition is important. Expanding on the specific components of culture media that influence MSC growth and functionality, and the challenges in formulating GMP-compliant media, would be insightful.

32. The impact of serum-free or chemically defined media on MSC characteristics, and how these changes might affect their therapeutic efficacy, warrants further exploration.

33. A concluding section summarizing the key challenges in immunological response management and MSC manufacturing, and suggesting future research directions or potential solutions, would provide a strong end to the manuscript.

Conclusions and Future Perspectives

34. The conclusion effectively summarizes the current state of cell-based therapies in neurodegenerative diseases (NDDs). However, it would be beneficial to briefly reiterate the specific challenges that have been encountered in translating preclinical successes into clinical treatments. This could include a summary of both the promising aspects and the hurdles like immune response, cell survival, and integration.

35. You mention that the molecular mechanisms underlying the beneficial effects of stem cells are still unidentified. Expanding on this point by suggesting potential research directions or methodologies that could help uncover these mechanisms would be insightful.

36. The section rightly identifies key challenges in translating stem cell therapies from bench to bedside, including optimal timing, dosage, and administration routes. Elaborating on these challenges with examples or potential solutions could provide a clearer picture for future research directions.

37. You touch upon the discrepancies between fresh culture-derived MSCs in preclinical studies and cryopreserved MSCs in clinical trials. A deeper discussion on how cryopreservation affects cell viability, functionality, and therapeutic efficacy would be beneficial. Additionally, discussing advancements in cryopreservation techniques and their potential to improve clinical outcomes would be valuable.

Comments on the Quality of English Language

The manuscript's quality of English language is generally good, demonstrating clarity and coherence in the presentation of complex scientific concepts. However, there are areas where the language could be refined to enhance readability and precision. A few suggestions include:

1. Some sentences are quite complex and could be broken down into shorter, more straightforward sentences. This would aid in easier comprehension, especially for readers who are not native English speakers or those not familiar with the technical jargon of the field.

2. While the passive voice is common in scientific writing, occasional use of the active voice can make the text more engaging and direct. Consider balancing the use of passive and active voices, especially in sections where you want to emphasize the actions or findings.

3. While the overall grammar is solid, a thorough proofreading could help catch minor grammatical errors and punctuation inconsistencies that may have been overlooked. Please find below a few comments and suggestions for improvement. It is advisable to review the entire content of the manuscript thoroughly for further refinements.

(1) "Although clinical translation is still at an early stage, several preclinical studies have been conducted to evaluate the neuroprotective mechanisms of stem cells in animal models."

Suggestion: Consider rephrasing for clarity, for example: "Despite being in the early stages of clinical translation, numerous preclinical studies have investigated the neuroprotective mechanisms of stem cells in animal models."

(2) "In this report, it was hypothesized that neuroprotective effect could be derived from the delivery of various neuro-trophic factors from newly formed blood vessels."

Correction: Change to "In this report, it was hypothesized that the neuroprotective effect could be derived from the delivery of various neurotrophic factors by newly formed blood vessels."

(3) "Moreover, there is evidence that the transplantation site itself also plays an important role."

Suggestion: This sentence is correct but could be more informative. For instance: "Additionally, evidence suggests that the site of transplantation plays a crucial role in the success of the therapy."

(4) "The therapeutic potential of BDNF for ALS has been recently investigated by Wang et al., who observed the beneficial role of BDNF-overexpressing hUC-MSC-derived motor neurons in the ALS model mice."

Correction: Revise for clarity and grammatical accuracy, for example: "Recently, Wang et al. investigated the therapeutic potential of BDNF in ALS. They observed that motor neurons derived from hUC-MSCs overexpressing BDNF had a beneficial role in ALS model mice."

(5) "In addition, recent advances in genome engineering and three-dimensional (3D) cell culture technology can provide new opportunities for human disease modeling and personalized treatments."

Suggestion: This sentence is grammatically correct but could be made more impactful. For example: "Furthermore, recent advancements in genome engineering and three-dimensional (3D) cell culture technologies are opening new avenues for modeling human diseases and developing personalized treatments."

Round 2

Reviewer 2 Report (New Reviewer)

Comments and Suggestions for Authors

I want to extend my heartfelt thanks to the authors for their meticulous and thorough revisions. The manuscript has significantly improved from its original submission, demonstrating the authors' earnest dedication to addressing all the issues I had highlighted earlier.

Comments on the Quality of English Language

Minor editing of English language required

This manuscript is a resubmission of an earlier submission. The following is a list of the peer review reports and author responses from that submission.

Round 1

Reviewer 1 Report

Comments and Suggestions for Authors

The manuscript entitled "Perinatal tissue-derived stem cells: an emerging therapeutic strategy for challenging neurodegenerative diseases " is a review manuscript where the authors have discussed the beneficial effects of use of perinatal stem cells (such as umbilical cord stem cells) as a potential treatment to brain diseases.

There are some major issues with this manuscript.

1. the authors have mentioned multiple times in the manuscript that the MSCs can become neurons and glial cells. MSCs are multipotent in nature and not pluripotent in nature to transdifferentiate in a different embryonic layer. Please make sure that the basics regarding the MSCs are written in a logical manner in the manuscript. Trans differentiation is unfortunately misused and most of times not analyzed properly in the current literature. The only proof of trans differentiation of MSCs into neuronal lineages should be investigated with electrophysiology studies, which are rarely done.

2. The authors work is poor on literature review. For example: the hypothesis of Fink et al 2013 (page 11) as stated by the authors is completely inaccurate. Please make sure that the manuscripts are cited correctly and cited for what they are stating.

3. The passage number at which the MSCs are transplanted plays a major role in the therapeutic outcome, please mention this for all the studies cited in this manuscript.

4. Please have sub heading between sections so that the text is clear and understandable to the readers.

Conclusions: The facts about the basics of stem cells, especially regarding MSCs are incorrect in this manuscript, moreover, the authors have mentioned the inaccurate facts from the cited papers. The manuscript in the current form should be rejected. 

Reviewer 2 Report

Comments and Suggestions for Authors

The authors summarized potential fetal stem cell therapies for neurodegenerative disorders. Also, outlined preclinical and clinical outcomes.

However, the comments below need to be addressed from a stem cell and regenerative medicine point of view.

1.     What are the potential ways to reduce immune rejection? A brief discussion would be good under a separate session for all the mentioned diseases.

2.     What are the cGMP-grade cryopreservation methods that need to be considered to meet future patient needs? It can be summarized as a table.

3.     Number of cells per injection and how many cell numbers? Make a table or schematic collectively for mentioning diseases.

4.     Methods to produce on a large scale to meet the clinical demands. A schematic would make a better understanding.
